# In vivo conformational space and defects of misfolded CFTR variants by covalent protein painting

Sandra Pankow ⬡ ✉, Tom Casimir Bamberger ⬡,
Salvador Martínez-Bartolomé ⬡, Robin Park & John R. Yates III ⬡ ✉

In vivo characterization of protein structures and structural changes after perturbation is still a major challenge and has impacted our understanding of the molecular events involved in protein misfolding diseases. To identify the true conformational space occupied by proteins in their native state in vivo, we recently developed a structural proteomics method named Covalent Protein Painting (CPP). Here, we show how CPP can be used to identify and quantify the conformational defects of proteins in the misfolding disease Cystic Fibrosis. We first report the discovery of a previously unreported opening mechanism for the Cystic Fibrosis Transmembrane Conductance Regulator (CFTR) as well as its conformational changes during biogenesis. Then we further reveal how misfolding of different CFTR variants in Cystic Fibrosis disturbs these conformational changes even upon treatment with current approved drugs and suggest possibilities to stabilize misfolded CFTR variants not or less responsive to these drugs such as N1303K CFTR.

The tertiary structure of a protein is closely linked to its function and determines how it interacts with other proteins. Changes to the tertiary structure, caused for example by mutations or environmental factors, are frequently associated with loss- or gain-of-function of a protein. Such changes result in a variety of protein misfolding diseases such as Cystic Fibrosis (CF), the most common genetic childhood disease in Caucasians, or Alzheimer's[1–3]. Determining and understanding such conformational defects presents a significant challenge to methods that produce high-resolution structures because of the unfavorable biophysical properties of many of the proteins that cause misfolding diseases. For example, many proteins are either unstable and thus difficult to produce and purify, contain long stretches of inherently disordered regions, are membrane proteins, or all of the above. Some of these challenges have been addressed by introducing stabilizing mutations and deleting inherently disordered regions, which was critical for the success of recent cryo-EM studies of the human Cystic Fibrosis Transmembrane Regulator (CFTR), an anion channel important for maintaining the salt balance in stratified epitheliums such as the lung and mutated in the misfolding disease Cystic Fibrosis. These studies provided invaluable insight into channel architecture and how the most prevalent mutation, ΔF508 CFTR, affects the protein structure[4–6]. However, the Cryo-EM structures can only reveal a static picture of the isolated and stabilized CFTR protein, which does not reflect the many conformational changes that can occur upon post-translational modifications (PTMs) such as PTM-dependent intramolecular domain interactions and interactions with other proteins and small molecules. At least 60 primary protein-protein interactions have been reported for CFTR and it is modified by over 50 different PTMs, several of which are located in a small disordered region, the RI element[7], which is deleted in recombinantly expressed CFTR used for the Cryo-EM studies. Recent studies using smFRET further revealed that alternate conformations of CFTR exist in the cell and have identified a critical role for the RI element in switching between a canonical state and an alternate state as well as defined an allosteric gating mechanism[8]. But even in these approaches, stabilizing mutations had to be introduced in addition to the smFRET labels, which could affect conformation in subtle ways as well.

Department of Integrative Structural & Computational Biology, The Scripps Research Institute, 10550 North Torrey Pines Road, La Jolla, CA 92037, USA.
✉e-mail: pankows@scripps.edu; jyates@scripps.edu

To identify and quantitatively characterize the large conformational space occupied by CFTR and misfolded CFTR mutants causing Cystic Fibrosis, we have developed Covalent Protein Painting (CPP), a structural proteomics approach that allows identification and quantification of protein misfolding events in vivo[9,10]. CPP covalently labels the epsilon amine of a lysine side chain in vivo. Lysine side chains contain primary amines that are positively charged at physiological pH. They occur primarily on exposed surfaces of the native tertiary structure of a protein where they are readily accessible for interaction with other proteins as well as for labeling with amine-reactive reagents[11]. If lysines are inside a protein structure, or are involved in a protein- or small molecule interaction, they are masked for labeling and therefore become solvent-excluded[12]. Thus, labeling reagents can be used to map protein structure and interactions by measuring the differences in the availability of the amino acid side chain to react with a label. In this work we show that CPP enables in vivo detection of CFTR conformational changes during biogenesis and activation, revealing a previously unrecognized opening mechanism. We further demonstrate that disease-causing CFTR variants disrupt these changes—even under approved therapies—and identify potential strategies to stabilize less responsive variants such as N1303K.

## Results

To characterize CFTR conformations in vivo, surface exposed lysines in human bronchial epithelial cells expressing wt CFTR (HBE16o-) were covalently labeled with isotope encoded dimethyl groups directly in dish for 10 min on ice according to the CPP workflow presented in Fig. 1A and as published previously[9]. Subsequently, cells were lysed and CFTR was co-immunoprecipitated using the CoPIT protocol[13]. Labeled CFTR and its interactors were then digested into peptides with lysine-insensitive chymotrypsin. Subsequently, we used a second isotope combination to label lysines that were solvent excluded in the cells in vivo but became accessible after digestion of the proteins into peptides. Using quantitative mass spectrometry, labeled lysines present in a complex cell lysate can be directly identified and quantified and their solvent accessibility determined.

Twenty-six lysine sites in CFTR were quantified in 4 different biological replicates, whereby distinct peptides containing the same lysine site can be present due to the nature of the chymotryptic digest and can occur in different charge states, each representing an individual measurement. Identified lysine sites were then mapped onto the CFTR Cryo-EM structure (PDB 5UAK)[4] showing a good general agreement between solvent accessibility determined by Cryo-EM and solvent accessibility determined by CPP (Fig. 1B, C; Table 1). For example, lysines located on the outside of the CFTR molecule such as Lys 283 were all >98% solvent accessible. In addition, CPP was able to quantify the solvent accessibility of lysines in intrinsically unstructured regions such as the Regulatory Insertion (RI) element and the Regulatory Region (R-region), which is not resolved in Cryo-EM structures.

We then performed CPP on isogenic CFBE41o- cells expressing ΔF508 CFTR and compared its solvent accessibilities with wt CFTR. Our experiment revealed large differences in solvent accessibility in the transmembrane helix 1 (TMH1) and the intracellular loop 2 (ICL2) as well as smaller, but significant differences in the RI element, the N-terminus of the nucleotide binding domain 1 (NBD1) and the NBD2 (Fig.1B, left panel). A label swap experiment confirmed the detected differences between wt and ΔF508 CFTR (Fig. 1B, right panel). Particularly interesting changes were observed for Lys 95 (K95) and Lys 464 (K464). Lys 95 (K95) is part of the narrow region of the internal vestibule and is fully accessible in wt CFTR and K464 is the Walker A lysine in NBD1, which is a non-hydrolytic ATP binding site, that was proposed to drive opening of phosphorylated CFTR channels by promoting the formation of NBD1:NBD2 dimers[14]. Both sites were less accessible in ΔF508 CFTR. Slight variations in quantification in the label swap experiment for each peptide are expected due to known factors

including isotope labeling–induced retention time shifts as well as inherent differences in sample preparation when samples are not simultaneously prepared and mass spectrometer performance. However, the largest difference was observed for K273 in the ICL2, which was solvent-accessible on average in only 60% of wt CFTR molecules, but to >95% in ΔF508 CFTR (Fig. 1D). The ICL2 is a short "coupling helix" that reaches into the NBD2 in what has been called a "ball-in-a-socket" motif[15]. K273 is located at the bottom of the loop and is solvent excluded in all published Cryo-EM structures (PBD 5UAK, 6MSM, 6O1V, 6O2P,7SVD,7SV, 8EIG, 7SVR, 8EJ1, 8EIO, 8FZQ)[4,5,8,16,17], although slight differences between wt CFTR and ΔF508 CFTR are seen in the size of the solvent accessible isosurface of K273. Solvent accessibility of K273 thus represents a CFTR conformation in which ICL2 does not reach into NBD2, e.g. an uncoupled state. Interestingly, 60 % of wt CFTR was also in such an uncoupled conformation.

To gain a better understanding of the observed differences between wt and ΔF508 CFTR and the difference in K273 solvent accessibility in vivo with the published cryo-EM structures, we performed CPP experiments on ΔF508 CFTR at permissive temperature of 28 °C, a temperature at which ΔF508 CFTR is efficiently trafficked to the plasma membrane and partially active[18] as well as on ΔF508 CFTR after treatment with the different FDA approved corrector drugs VX-809, VX-445 (elexacaftor) and VX-661 (tezacaftor) or the triple combination of VX-770, VX-445 and VX-661 (Trikafta), which is the most effective treatment to date for CF[19–21]. Treatment with VX-809 did not significantly change solvent accessibility of detected lysines in ΔF508 CFTR except for lysine 1218, which is located in NBD2, is ubiquitinated in ΔF508 CFTR and has been implicated in lysosomal degradation of ΔF508 CFTR[7,22]. Likewise, treatment with the two correctors VX-661 and VX-445 did not induce significant lysine-sensitive conformational changes (Fig.2A, B). However, Trikafta treatment increased coupling of the ICL2 loop with NBD2 by 15 % as shown by reduced solvent accessibility of Lys 273 (81.5% solvent accessible) (Fig. 2B). The reduction in solvent accessibility was similar and even slightly larger than that observed for Lys 273 at permissive temperature of 28 °C (89%), showing that coupling of ICL2 to NBD2 occurred in about 11% of CFTR molecules at permissive temperature and in 19% of CFTR molecules upon Trikafta treatment (Fig. 2B). We therefore speculated that ICL2 coupling to NBD2 may be crucial for CFTR activity.

Comparing solvent accessibility of K273 in Cryo-EM models of inactive CFTR[4] (PDB 5UAK) and activated, dimerized CFTR[6] (PDB 6MSM) revealed that K273 was solvent excluded in both models as well as in additional homology models[23]. To investigate the possibility that in vivo activation may be different from in vitro activation of CFTR, we treated HBE cells with 500 nM VX-770 for 10 min to stimulate wt CFTR activity[24] and then measured K273 accessibility by CPP. VX-770 is a channel potentiator that opens CFTR channels independent of ATP hydrolysis and is used to treat CF caused by the G551D mutation[25–27]. In contrast to non-stimulated wt CFTR, in which K273 is solvent accessible in 60% of molecules, activation of wt CFTR by VX-770 treatment led to complete solvent exclusion of K273. Only in 0.66 % of CFTR molecules was K273 still solvent accessible, thus showing a clear correlation of K273 accessibility with wt CFTR activity (Fig. 2C). The coupling of ICL2 to NBD2 upon activation with VX-770 was accompanied by additional surface accessibility changes in the vestibule (K95), the chloride pathway (Lys 370) and the Walker A lysine Lys 464, all of which became less accessible upon treatment with VX-770 (Fig. 2D, Supplementary Fig. 1A). The data suggested that the CFTR conformation in which K273 is solvent excluded and the ICL2 reaches into NBD2 is an active open state, while the discovered conformation in which K273 is solvent accessible represents an inactive state in which ICL2 is uncoupled from NBD2.

To better understand how Trikafta treatment leads to activation of this ΔF508 CFTR conformation, we further compared the solvent accessibility of lysines in ΔF508 CFTR upon Trikafta treatment with

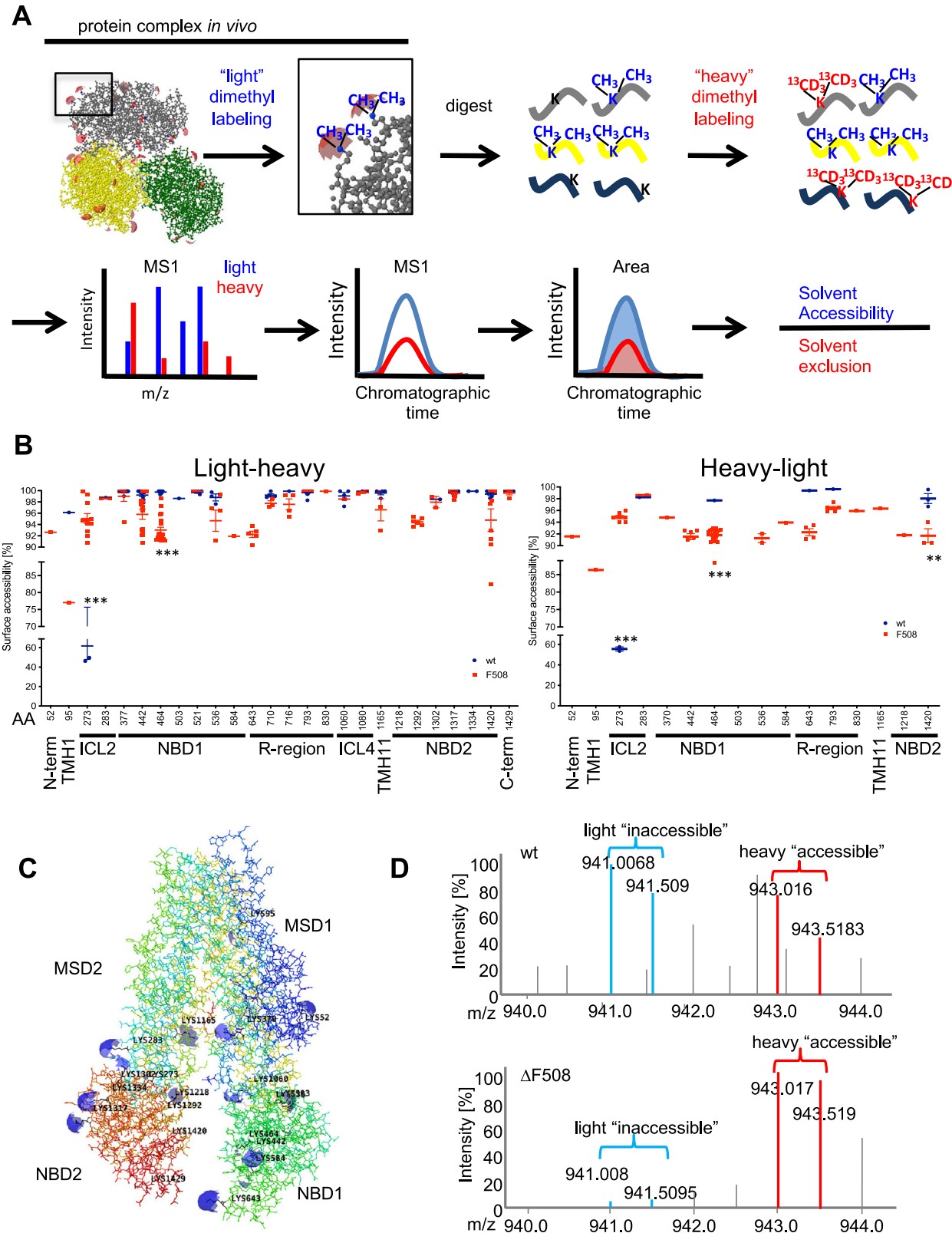

that of ΔF508, activated wt and wt CFTR. Interactions at the NBD transmembrane domain interfaces, e.g. NBD1 with ICL1/ICL4 and NBD2 with ICL2 and ICL3, are responsible for coupling ATP binding and hydrolysis to channel gating by transmission of conformational changes from the NBDs to the TMDs[28]. Upon Trikafta treatment K1060 located in ICL4 also became less solvent accessible in F508 CFTR, suggesting that Trikafta led to coupling of ICL4 with NBD1 and restores

the disturbed ICL4/NBD1 interaction in ΔF508 CFTR to some degree. Thus, Trikafta restored the transmission of the NBD conformational changes to the TMDs in ΔF508 CFTR- at least partially. Furthermore, Trikafta treatment also induced a large conformational change at K643 that is also seen in activated wt CFTR as well at K716, also in the R-region, confirming that conformational changes to the R-region play an important role in channel activation. Interestingly, a small, but

**Fig. 1 | CPP analysis of wt and ΔF508 CFTR. A** Experimental and mass spectrometric workflow. **B** Solvent accessibility of quantified lysines in different CFTR domains upon labeling with $CH_3$ ("light") followed by $^{13}CD_3$ ("heavy") labeling (left panel) and upon label-swap (right panel). Individual points reflect independent measurements of the solvent accessibility for each lysine site and detected peptide. As more than one peptide may contain a specific lysine site and may occur in more than one charge state, more than one measurement per biological replicate may be obtained. Statistical significance was assessed with PCQ (One-Way ANOVA) and

Benjamini-Yekuteli-Krieger post correction. ***p-values ≤ 0.0001, **p-values ≤ 0.0002. **C** Quantified lysines were mapped onto the CFTR Cryo-EM structure (PDB:5UAK) and solvent accessible isosurfaces displayed as blue spheres.
**D** Representative MS1 spectra of K273 depicting light and heavy precursors used for quantification in wt CFTR (upper panel) or ΔF508 CFTR (lower panel). Panel (**B**) left side: Wt CFTR n = 4; ΔF508 CFTR n = 4 (biological replicates). Error bars reflect standard error of measurement, the mean is indicated by a line. For the label swap experiment in B (right side): wt CFTR n = 2, ΔF508 CFTR n = 2 (biological replicates).

## Table 1 | Mapping and comparison of lysine solvent accessibilities to Cryo-EM structures

| Position | Sequence | Mean surface accessibility (%) | | Wt Cryo-EM structure | Wt Cryo-EM structure, ATP bound, phosphorylated |
|---|---|---|---|---|---|
| | | wt | ΔF508 | | |
| K52 | QIPSVDSADNLSEKL | nd | 92.6 | accessible (93.75 Å$^2$) | accessible (13.39 Å$^2$) |
| K95 | LGEVTKAVQPL | 96.1 | 77.0 | partially accessible (3.51 Å$^2$) | accessible (35.78 Å$^2$) |
| K273 | VITSEMIENIQSVKAY | 61.8 | 95.0 | inaccessible (0.48 Å$^2$) | partially accessible (6.92 Å$^2$) |
| K283 | CWEEAMEKMIENLRQTEL | 98.7 | 98.9 | accessible (53.98 Å$^2$) | accessible (39.15 Å$^2$) |
| K370 | DSLGAINKIQDF | 99.8 | 99.0 | accessible (41.45 Å$^2$) | accessible (29.92 Å$^2$) |
| K370/K377 | DSGAINKIQDFLQKQEY | 99.8 | 98.9 | accessible (41.45 Å$^2$, 39.11 Å$^2$) | accessible (29.92 Å$^2$, 36.11 Å$^2$) |
| K411 | not present in CryoEM | | | | |
| K420 | not present in CryoEM | | | | |
| K442 | SLLGTPVLKDINF | 99.2 | 95.7 | accessible (39.53 Å$^2$) | accessible (47.84 Å$^2$) |
| K464 | AVAGSTGAGKTSLL | 99.7 | 93.0 | accessible (9.59 Å$^2$) | partially accessible (4.36 Å$^2$) |
| K536 | AEKDNIVLGEGGITL | 98.8 | 94.7 | accessible (51.94 Å$^2$) | accessible (26.99 Å$^2$) |
| K584 | GYLDVLTEKEIF | nd | 91.9 | accessible (8.91 Å$^2$) | accessible (18.03 Å$^2$) |
| K1060 | ESEGRSPIFTHLVTSLKGLW | 99.0 | 98.5 | accessible (49.05 Å$^2$) | partially accessible (3.15 Å$^2$) |
| K1080 | FETLFHKAL | 99.9 | 99.7 | accessible (59.33 Å$^2$) | accessible (35.59 Å$^2$) |
| K1165 | KFIDMPTEGKPT | 99.5 | 96.6 | accessible (19.66 Å$^2$) | accessible (54.57 Å$^2$) |
| K1218 | TAKYTEGGNAILENISF | nd | 89.3 | accessible (41.39 Å$^2$) | accessible (30.56 Å$^2$) |
| K1292 | GVIPQKVF | nd | 94.4 | accessible (48.20 Å$^2$) | partially accessible (3.01 Å$^2$) |
| K1302 | RKNLDPYEQWSDQEIW | 98.4 | 97.9 | accessible (26.64 Å$^2$) | accessible (13.86 Å$^2$) |
| K1317 | KVADEVGLRSVIEQFPGKLDF | 99.7 | 99.5 | accessible (37.29 Å$^2$) | accessible (53.09 Å$^2$) |
| K1334 | RSVIEQFPGKLDF | 99.9 | nd | accessible (56.92 Å$^2$) | accessible (49.60 Å$^2$) |
| K1420 | LVIEENKVRQY | 99.3 | 94.8 | accessible (51.71 Å$^2$) | accessible (51.65 Å$^2$) |
| K1429 | DSIQKLL | 99.9 | 99.5 | accessible (57.27 Å$^2$) | accessible (33.64 Å$^2$) |

Quantified lysines and respective peptide sequences were mapped onto the available Cryo-EM structures 5UAK and 6MSM and solvent accessible isosurfaces probed with a rolling probe.

significant change was also observed for the very C-terminus both in activated wt CFTR as well as Trikafta treated ΔF508 CFTR. However, the coupling of the ICLs with the TMDs and R-region changes were not accompanied by the same changes to residues K464 or K370 involved in channel gating and the ion permeation pathway as in wt CFTR. Instead, solvent accessibility remained similar to that of untreated or solvent control treated ΔF508 CFTR (Fig. 2D). Furthermore, changes to the N-terminus as indicated by altered solvent accessibility of K52 in activated wt CFTR were not observed in Trikafta treated ΔF508 CFTR. Thus, this analysis revealed similarities between wt CFTR and ΔF508 CFTR activation as well as differences that show that the activated ΔF508 CFTR channel is in a slightly different conformation than the activated wt CFTR channel in vivo.

To probe whether the previously undescribed inactive CFTR conformation is associated with gating of the channel or is rather an inactive confirmation attained by CFTR when it is not present at the plasma membrane, we performed additional CPP experiments with CFTR variants N1303K CFTR and G551D CFTR. N1303K CFTR, like ΔF508 CFTR, is misfolded and not present at the plasma membrane, while G551D CFTR is trafficked normally to the plasma membrane, but has a gating defect[29–32]. CPP results on these CFTR variants showed that

solvent availability of K273 was high (>95%) in the mainly ER and Golgi located, misfolded CFTR variants N1303K and ΔF508 CFTR, but that it was solvent excluded in G551D, suggesting that the observed inactive conformation is associated with immature CFTR that is not present at the plasma membrane (Fig. 3A, B). Therefore, we generated a K273A CFTR point mutant, to prevent insertion of the ICL-2 loop into the NBD2 and analyzed biogenesis of K273A CFTR by confocal microscopy. Confocal images showed that the K273A CFTR mutant accumulates in the ER and Golgi based on co-localization with the ER and Golgi detection probes (Fig. 4A, Supplementary Fig. 2). Staining with autophagy markers SQSTM and cleaved LC-3 remained negative as expected (Supplementary Fig. 3). To better understand the importance of K273, we performed an interactome analysis of K273A, wt and ΔF508 CFTR and GFP controls[33]. Gene Ontology (GO) enrichment analysis revealed that proteins localized to membrane bounded organelles and vesicles were highly enriched as interactors of K273A CFTR as well as proteins involved in endosomal transport and vesicle mediated transport (Fig. 4B, C, Supplementary Data 1). Examples of these interactors include the ESCRT complex, and the retromer cargo-selective complex (CSC), specifically the SNX3-retromer. The ESCRT complex is required for endosomal sorting of protein cargo into

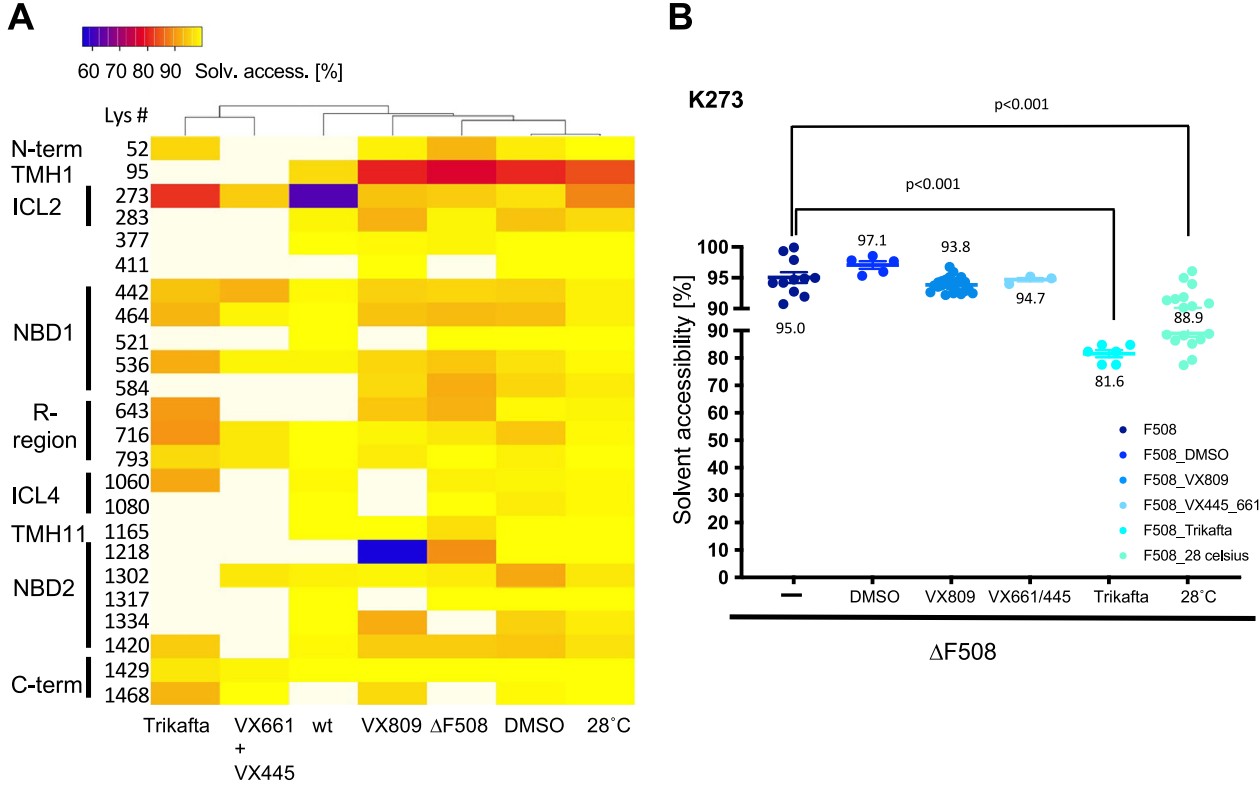

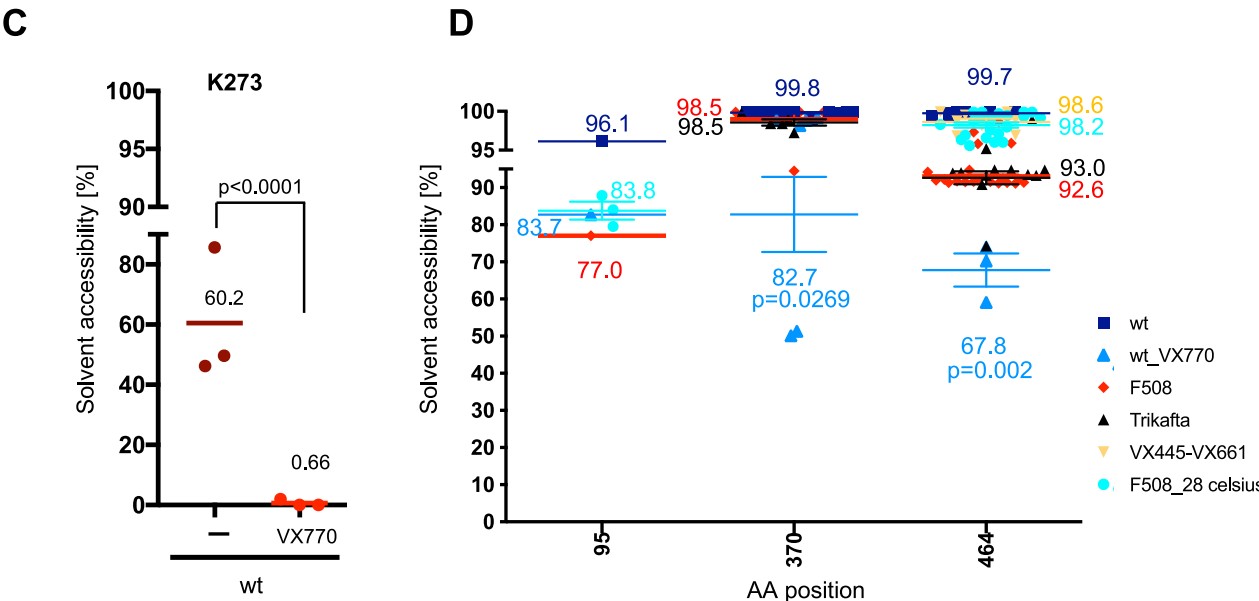

**Fig. 2 | Conformational changes upon correction and activation of the CFTR channel with correctors and activators. A** Heatmap displaying solvent accessibility changes of ΔF508 CFTR upon treatment of CFBE41o- cells with Trikafta (24 h), the two corrector compounds VX-661(5 μM, 24 h) and VX-445 (3 μM, 24 h), corrector VX-809 (5 μM, 36 h), at permissive temperature of 28 °C (24 h), or in DMSO solvent controls, untreated ΔF508 CFTR and of wt CFTR. **B** Scatter plot displaying solvent accessibility changes of K273 in the ICL-2 loop of ΔF508 CFTR upon treatment with Trikafta, correctors VX-661, VX-445, VX-809 and at permissive temperature as described above. Mean is indicated for each condition. **C** K273 solvent accessibility changes upon

activation of the wt CFTR channel in HBE41o- cells treated with VX-770 (500 nM, 10 min). **D** Changes in solvent accessibility of lysines located in the ion permeation pathway and ATP binding site upon CFTR channel activation and correction in cells treated with VX-770, Trikafta, correctors, or at permissive temperature as described in A. Wt (*n* = 4), wt_VX770 (*n* = 3), ΔF508 (*n* = 4), ΔF508_VX809 (*n* = 3), ΔF508_DMSO (*n* = 3), ΔF508_28 °C (*n* = 3), ΔF508_Trikafta (*n* = 3), ΔF508_VX661/VX445 (*n* = 3). n represents number of biological replicates. Statistical significance was assessed by One-Way ANOVA with Benjamini-Hochberg post-correction. Error bars reflect standard error of measurement, the mean is indicated by a line.

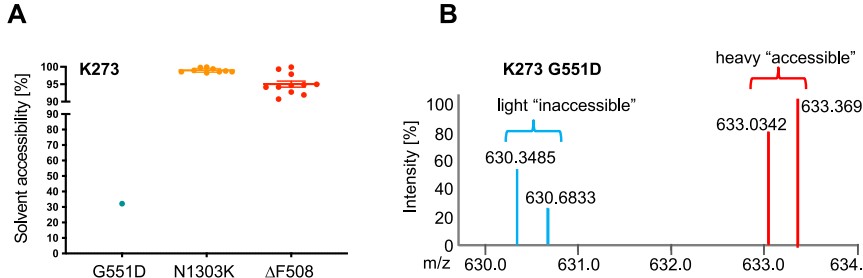

**Fig. 3 | K273 solvent accessibility in G551D and N1303K CFTR. A** Solvent accessibility of K273. **B** Representative MS1 spectrum of the extracted precursor ions of the peptide containing K273 in G551D CFTR. Light and heavy precursors and measured m/z for each are indicated. G551D (*n* = 3), N1303K (*n* = 3). n represents biological replicates. Error bars reflect standard error of measurement, the mean is indicated by a line.

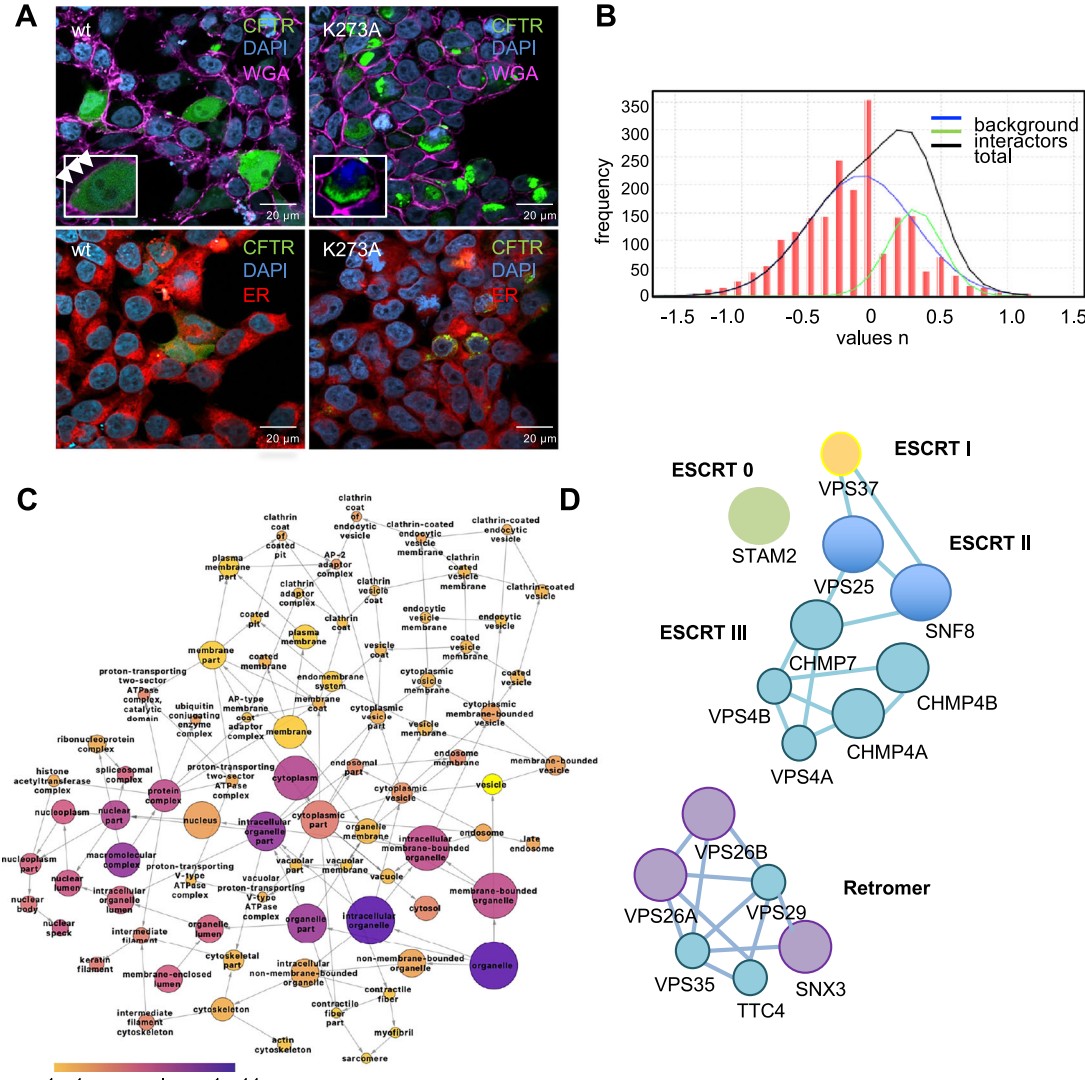

**Fig. 4 | K273A CFTR interactome and cellular localization. A** HEK293T cells transiently transfected with GFP tagged wt or K273A CFTR (green) were stained with wheat germ agglutinin (pink) to visualize the plasma membrane (upper panel). ER was stained with the ER Cytopainter kit (lower panel). Nuclei were counterstained with DAPI (blue). *n* = 3 biological replicates. Inserts show magnified regions of the image. White arrows indicate localization of GFP-tagged CFTR at the plasma membrane. **B** Interactome analysis of K273A CFTR. True K273A CFTR interactors were distinguished from background using CoPITgenerator by comparing the interactome to that of wt CFTR, ΔF508 CFTR and controls (GFP, CFTR null cells). **C** Network of enriched GO-terms in the K273A CFTR interactome. The area of a node is proportional to the number of genes in the interactome data set annotated to the corresponding GO category. Corrected *P*-values for statistical overrepresentation of a GO category in the dataset are indicated by color according to the color scalebar. Arrows indicate hierarchy of the ontology terms. **D** ESCRT and retromer complexes were specifically enriched in the K273A interactome as determined by BiNGO, and annotated interactions in the String database between the identified interactors are displayed.

multivesicular bodies (MVBs)[34,35], and the SNX3-retromer mediates retrograde transport from the endosome to the trans-Golgi-network (TGN), and transport from endosome to plasma membrane. It was also previously shown to promote cell surface expression of ENAC sodium channels[36–38] (Fig. 4D). Most of these proteins also play pivotal roles in the interactome of ΔF508 CFTR under different conditions[33].

## Discussion

Here, we report the conformational space of CFTR and misfolded CFTR variants in vivo and how these differ from previously published structures. The data suggest previously unknown conformational rearrangements that happen upon activation of the channel in the ion permeation pathway for both wt and ΔF508 CFTR and upon Trikafta treatment, which were not observable by Cryo-EM[5]. In particular, we report the discovery of a CFTR conformation that reflects an immature, closed state of the channel. This conformation is attained by 60 % of wt CFTR molecules in the cell as well as the misfolding mutants ΔF508 CFTR and N1303K. The most striking feature of the discovered immature CFTR conformation is the altered solvent accessibility of K273 in the ICL2 coupling helix. In general, ABC transporter ICLs mediate the coupling of ATPase (catalytic) activity at the nucleotide binding sites to substrate translocation through the membrane spanning domains (MSDs)[39]. Previous data suggested that ICL2 is involved in stabilization of the full conductance state of the CFTR Cl channel, as a 19 amino acid deletion mutant of CFTR ICL2 is non-responsive to forskolin stimulation and does not reach the cell surface[40]. In models of the ABC transporter P-glycoprotein the ICLs may move like a ball in a socket to accommodate conformational change during the gating cycle[15]. In this arrangement, the ICL2 loop reaches into the NBD2 as in all current Cryo-EM structures of CFTR and thus is not solvent accessible. Consistent with those data, such solvent exclusion of K273 was observed upon activation of the CFTR channel with VX-770. Interestingly, crosslinking cysteines engineered in CFTR's ICL2 and NBD2 decreased the open probability of the channel ($P_o$) dramatically[41,42], suggesting that movement of the ICL2 is needed for proper channel gating. However, if K273 is fully solvent accessible as it is in ΔF508, N1303K and 60% of wt and G551D CFTR, it must represent a conformation in which the ICL2 is removed from the socket in NBD2 (i.e., an "uncoupled" state), and thus represents an inactive CFTR conformation.

One could interpret the different conformations simply as open/closed states during the gating cycle while CFTR is located at the plasma membrane. However, the fact that this inactive conformation is also observed in ΔF508 and N1303K CFTR, two CFTR variants that are misfolded and are largely degraded before reaching the plasma membrane, suggests otherwise. Furthermore, based on its interactome and confocal images, K273A mutation led to accumulation of CFTR in the ER, Golgi and Trans-Golgi network (TGN), ERGIC and endosomes. This indicates that the inactive conformation is an immature CFTR conformation that is attained during transport from the ER to the plasma membrane and may prevent accidental channel activation during biogenesis and transport. Thus, CPP discovered an immature CFTR conformation that was previously unobservable. It is possible that this conformation is stabilized by interactors that prevent ICL2 from reaching into NBD2 or by internal molecular re-arrangements, such as movement of the R-domain or the RI element upon phosphorylation. While movement of the R-domain from its place between the two NBDs to the peripheral surface has been observed in the PKA phosphorylated, ATP-bound CFTR[43], it cannot be the only factor that prevents the coupling of ICL2 to NBD2, as ICL2 also reaches into NBD2 in the structure of the un-phosphorylated, ATP-free CFTR. We therefore suspect that additional in vivo phosphorylation events are needed, or that movement of the RI element (which is deleted in all CFTR structures) is involved since it is highly phosphorylated in wt CFTR, but not in ΔF508 CFTR and the amount of RI element

phosphorylation correlates with the amount of fully mature CFTR in the plasma membrane[7]. Stabilization by other interactors would also explain why this conformation was not observed in Cryo-EM studies, where such interactors are necessarily absent, and why no further conformational change was seen upon treatment of ΔF508 CFTR with VX-770 compared to elexacaftor/tezacaftor treated ΔF508 CFTR[5]. The change in ICL2 conformation is preserved across different cell lines showing the generalization of the result and its likely importance for CFTR function. In addition, CFTR conformation also appears to be very similar between different cell lines and conditions for the majority of residues (Supplementary Table 1, Source Data) - agreeing with the vast body of literature that show preserved CFTR function, temperature sensitivity and turnover in FRT, CFBE41o, HBE41o- and HEK cells, all of which have been widely used to study CFTR function and turnover[44–46].

The CPP results also present an opportunity to further examine changes in the ion permeation pathway that occur upon channel activation. Changes include the walker A lysine K464 becoming less solvent accessible upon VX-770 activation, likely reflecting ATP-binding and dimerization of the NBDs. This is consistent with the observation that NBD binding symmetrically closes off the ATP-binding sites in the activated state[43]. Further changes include K95 and K370, two charged residues in the vestibule and close to the chloride entry pore[4], which become partially inaccessible in activated wt CFTR treated with VX-770 and the G551D mutant. This could reflect a percentage of channels that have closed, or may reflect conformational changes that reflect regional pore constriction/tightening at the mouth of the pore and the vestibule. Regional pore constriction/tightening is consistent with a role of these positively charged residues in anion selection and in creating an electrostatic potential for chloride ions to move along the pore[47]. Comparing CPP results across all analyzed CFTR mutants and upon activation, we also noticed changes in solvent accessibility of K52, which is part of the "lasso motif" and a structure unique to CFTR[48]. K52 was accessible in ΔF508 and N1303K, but only partially accessible in G551D and in VX-770 activated wt CFTR, suggesting a role of the lasso in channel activation or conformational changes occurring upon localization to the plasma membrane. Such a role would be consistent with previous observations that the lasso motif is important for interactions with the membrane traffic machinery and would confirm suggestions that the lasso regulates CFTR channel gating through interactions with the R domain[43,49–51]. The CPP data revealed a few changes that appear to be unique to N1303K, ΔF508 and G551D CFTR and appear to be related to the misfolding and the gating defect, respectively. In comparison to ΔF508 CFTR, N1303K CFTR exhibited higher solvent accessibility at K584 in NBD1 and K643 located at the border of the NBD1 and R-region. The G551D mutant exhibited a specific difference in solvent accessibility of K536. A change unique to ΔF508 CFTR was the slightly altered solvent accessibility of K1420 in NBD2. Interestingly, structural differences between wt and ΔF508 CFTR in NBD2 have been previously identified by limited trypsin digestion assays[52,53].

Finally, CPP data also allowed us to better rationalize a VX-809 corrector mechanism. VX-809 has been reported to bind to NBD1 using isolated NBD1 and to stabilize domain:domain interfaces, particularly NBD1:ICL4, although others observed that VX-809 alone does not stabilize isolated NBD1[54–56]. In contrast, Cryo-EM structures revealed a binding site formed by TM1, 2, 3, and 6[17]. CPP revealed in addition a reduction in solvent accessibility of K1218 by >30%, a very clear change that was not observed in any other condition (Source Data file). Interestingly, treatment of ΔF508 CFTR with VX-809 has been shown to decrease the proteolytic sensitivity of NBD2, indicating a more compact conformation of ΔF508 NBD2, which would be in line with a reduced solvent accessibility of K1218[19]. Furthermore, point mutation of K1218 to K1218R has been shown to increase the amount of CFTR, stabilize it at the plasma membrane and reduce lysosomal degradation of CFTR[22]. Deletion of K1218 also improves folding

efficiency of ΔF508 CFTR[53]. Thus, binding of VX-809 to the region or K1218 itself might induce the observed more compact conformation of NBD2 in vivo and could diminish ΔF508 CFTR targeting to the lysosome, although it is unlikely to completely prevent it, as multiple mechanism exist for peripheral quality control of CFTR[57,58]. Such a mechanism would agree with the moderate rescue of ΔF508 CFTR processing observed for VX-809.

## Methods

### Cell culture, plasmids and point mutations

CFBE and HBE cells were grown in A-MEM supplemented with 10% FBS, 1% Penicillin-Streptomycin (GibCo, Carlsbad, CA) and appropriate antibiotics at 37 °C, 5% CO$_2$. FRT cells expressing either wt CFTR or the variants G551D, N1303K or ΔF508 CFTR were cultured in Ham's F12 medium, Coon's modification supplemented with 5 % FBS and 100 μg/ml Hygromycin. K273A CFTR was generated using the Quick change method (Quiagen) in the pCMV6-ac-CFTR-GFP plasmid (Origene). Hek293T cells were transfected with Lipofectamine 3 according to manufacturer's recommendation (Thermo Fisher, Carlsbad, CA) and expression of K273A CFTR was analyzed 48 h to 72 h after transfection.

### In vivo dimethyl labeling and CFTR-IP

Cell culture media was removed and the intact cell layer was washed with 1x PBS before labeling of surface-exposed lysines with either formaldehyde (CH3, 0.3 % final concentration) and sodiumcyanoborohydrate (NaBH3CN, final concentration 30 mM) for light labeling or with deuterated formaldehyde containing $^{13}$C ($^{13}$CD3) and deuterated sodiumcyanoborohydrate (NaBD$_3$CN) for heavy labeling. While the labeling reaction is complete in <20 s[9], the labeling reaction was allowed to continue for 10 min on ice for practical reasons before it was stopped by adding 1% ammonium acetate (NH$_4$C$_2$H$_3$O). Subsequently, cells were harvested with a cell scraper and lysed by adding 2 x TNI buffer[13] containing complete Ultra protease inhibitor (Roche) and Halt phosphatase inhibitor (Pierce). The cell lysate was sonicated for 3 min in a water bath sonicator and insoluble material was removed by centrifugation at 18,000 x g, 4 °C, 15 min. CFTR-IP was then performed as described in Pankow et al.[13], except that the last three washes were performed with HNN buffer (50 mM Hepes, pH 7.5, 150 mM NaCl, 1 mM EDTA). Proteins were precipitated by methanol-chloroform precipitation (Lysate: Methanol: Chloroform, 1:4:1, v:v:v) and the precipitate was washed with 95% Methanol before digestion. For interactome analyses, GFP controls and K273A CFTR were immunoprecipitated using GFP-Trap Agarose (Chromotek) and interactomes were analyzed and compared to the interactomes of wt CFTR, ΔF508 CFTR and CFTR null cells with CoPITgenerator as described previously[13,33]. Interactors were filtered for confidence with a P-value of >0.9 in CoPIT generator[13,33]. GO Enrichment analysis of identified interactors was carried out using GO Miner[59] and a network of statistically overrepresented GO categories in the K273A CFTR interactome was determined and created using Cytoscape 3.2.8. and BiNGO using corrected p-values as described in Maere et al.[60]. Interactions between identified interactors in the ESCRT complexes and the retromer complex were identified and visualized with the GeneMANIA 2.2 Plugin in Cytoscape 3.2.8 using physical interactions reported in BioGRID-small scale studies, BIOGRID and BIND as well as Pathway information in Pathway Commons[61].

### Protein digestion and peptide labeling

Precipitated proteins were resolubilized by sonication in 0.2% Rapigest, 100 mM HEPES, pH 8.0 in a water bath sonicator. The protein lysate was heated to 95 °C for 10 min, reduced by 5 mM TCEP (Pierce), and alkylated with 10 mM iodoacetamide (Sigma-Aldrich, St.Louis, MO) before digestion with chymotrypsin (Protea Biosciences, Morgantown, WV, 1:100 w:w) at 30 °C for 14–16 h. Peptides were labeled according to Boersema et al. ref. 62 with 0.16% CH3 or $^{13}$CD3 and 30 mM NaBH3CN or NaBD$_3$CN, respectively, for 1 h at room temperature. The labeling reaction was stopped with 1% NH$_4$C$_2$H$_3$O, Rapigest was inactivated by incubation with 9% formic acid (1 h, 37 °C) and samples were reduced to near dryness in vacuo. Tryptic digestion of unlabeled CFTR IPs was carried out as described previously[13].

### Mass spectrometry

Peptides were reconstituted in buffer A (95% H$_2$O, 5% acetonitrile/0.1% formic acid), loaded onto a preparative MudPIT column and analyzed by nano-ESI-LC/LC-MS/MS on an LTQ-Elite (Thermo Fisher, San Jose, CA) by placing the triphasic MudPIT column in-line with an Agilent 1200 quaternary HPLC pump (Agilent, Palo Alto, CA) and separating the peptides in multiple dimensions with a 10-step gradient (0%, 10%, 20%, 30%, 40%, 50%, 60%, 70 %, 80%, 90% Buffer C (500 mM ammonium acetate/5% acetonitrile/0.1% formic acid)) over 20 h as described previously[63]. To avoid cross contaminations between different samples, each sample was loaded onto a fresh column. Each full scan mass spectrum (400−2000 m/z) was acquired at 60,000 R, followed by 20 data-dependent MS/MS scans at 35% normalized collisional energy and an ion count threshold of 1000. Dynamic exclusion was used with an exclusion list of 500, repeat time of 60 s and asymmetric exclusion window of -0.51 and +1.5 Da. Alternatively, peptides were separated with an 80 min pre-formed gradient on Evosep One tips and analyzed with a TimsTOF Pro (Bruker) in data-dependent mode. Monoisotopic peaks were extracted from Raw-files with RawConverter[64] and MS/MS spectra were searched with ProLuCID[65] against the human Uniprot database (release Nov 2019) using a target-decoy approach in which each protein sequence is reversed and concatenated to the normal database[66]. The search was carried out using a search window of 50 ppm, no enzyme specificity, N-terminal static modification according to protein dimethyl label (e.g. 28.0313 or 36.0757), and lysine dimethylation (28.0313) and carbamidomethylation of cysteine (57.02146) as static modifications. Additionally, oxidation of methionine (15.9949) and serine, threonine or tyrosine phosphorylation (79.9663) were considered as differential modifications. A lysine mass shift of 8.0444 was used for metabolic labeling search. Minimum peptide length was set to 7 amino acids (AA). Search results were filtered to <0.2% FDR on spectrum level, corresponding to <0.5% peptide level FDR using DTASelect 2.1 within the Integrated Proteomics Pipeline (IPA, San Diego, CA). Additionally, a deltaCN value of ≥ 0.1, and precursor delta mass cutoff of <8 ppm were applied as filter. As it is not uncommon for certain peptides−particularly those from low-abundance proteins or with challenging ionization properties−to be identified in only a subset of replicates, mass spectra for such peptides were manually validated.

### Quantification of solvent-excluded vs solvent-exposed lysines and mapping of solvent accessible areas

To determine the proportion of solvent-exclusion to solvent-exposure of a lysine residue, the area under each peptide peak was quantified using Census and the Integrated Proteomics Pipeline (IPA, San Diego, CA) as described previously[67,68]. Individual isotopes were extracted within a 10 ppm window. A determinant factor of ≥ 0.5, a profile score of ≥ 0.5 and no retention time shift allowed were set as parameters to filter for correct peak assignments. The determined ratio of light over heavy directly reflects solvent-exposure/ solvent-exclusion. Reported CFTR peptide ratios were further manually verified and precursor identity, peak shape and signal quality manually assessed. Protein structures available in PDB were visualized with JMOL (version 14.30.1) and the isosurface (isoSurface saSurface) of the epsilon amino group of lysine calculated and visualized in JMOL with a rolling probe radius of 1.2 Å. Area ratios were expressed as percentage of surface accessibility as described[9] and plotted in Prism 7 (Graphpad Inc.) Statistical analysis of area ratios per site was carried out in PCQ as described[69]. Briefly, ratio values for each lysine site were calculated with the SoPaX algorithm, which is part of ProteinClusterQuant (PCQ, https://github.

com/proteomicsyates/ProteinClusterQuant), a One-way ANOVA analysis carried out and obtained results subjected to Benjamini-Krieger-Yekuteli correction.

## Immunofluorescence staining and western blotting

Cells were grown in Lab Tek II chambers (Thermo Fisher), fixed with 4 % paraformaldehyde/1xPBS for 10 min at RT and solubilized with 0.1% Triton X-100 /1xPBS before staining with antibodies against LC3A/B (Cell Signaling Technologies, cat no. 12741S) and SQSTM1 (Cell Signaling Technologies, cat no. 8025S). The plasma membrane was visualized by wheat germ agglutin CF 594 conjugate (Biotium) and the ER was visualized using the ER Cytopainter kit (Abcam) according to manufacturer's recommendations. Slides were mounted in ProLong Gold Antifade Mount with DAPI to visualize nuclei (Thermo Fisher). Images were taken with a Zeiss LSM 710 confocal laser scanning microscope. For live cell staining, cells were incubated for 30 min with Golgi Detection Probe working solution prepared in phenol-free DMEM media, washed twice with phenol-free media according to the manufacturer's instructions (BioLegend, cat no. 421908) and imaged with a Zeiss Axioscope inverted microscope immediately after. Protein lysates were prepared in TNI buffer as described previously[13]. CFTR was detected using mouse monoclonal antibodies 24.1 (ATCC) and M3A7 (EMD Millipore). Antibodies against b-actin (AC-15, Sigma) and Na + /K+ATPase a (H300, sc28800, Santa Cruz) were used for normalization. Horseradish-peroxidase-conjugated secondary antibodies (Jackson Immunoresearch) were detected with enhanced chemiluminescence (ECL, Pierce).

## Reporting summary

Further information on research design is available in the Nature Portfolio Reporting Summary linked to this article.

# Data availability

The datasets generated in this study are available at the MassIVE repository under accession number MSV000096490 and will be made public upon publication. Source data are provided with this manuscript. Source data are provided with this paper.

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

## Acknowledgements

We would like to thank E. Sorscher (University of Alabama) for the kind gift of the FRT cell lines and Claire Delahunty (TSRI) for reading the manuscript and editing it for clarity. Funding was provided by NIH grants R01HL131697-05, R01HL165168 and R33CA212973-01 awarded to John R. Yates III.

## Author contributions

S.P., C.B. and J.R.Y. designed the research and S.P. and C.B. performed the experiments. S.M.B. implemented the protein residue-specific quantification and SoPaX in PCQ. R.P. supported mass spectrometric data analysis and developed quantification software. J.R.Y. provided materials and funding. S.P. wrote the manuscript and prepared the figures with help from all authors. All authors read and approved the manuscript.

## Competing interests

The authors declare no competing interests.
