## [Transparent Peer review file · Nature Communications]

In vivo conformational space and defects of misfolded CFTR variants by Covalent Protein Painting

Corresponding Author: Dr Sandra Pankow

Version 0:

Reviewer comments:

Reviewer #1

(Remarks to the Author)

In this study, the authors interrogate the in vivo conformational changes of four CFTR variants (wild type [WT], Δ F508, G551D and N1303K) by utilizing the previously developed and published Covalent Protein Printing (CPP) method. The CPP is based on the determining the extent of covalent modification of exposed epsilon amines of Lys residues by MS quantification that was applied here for probing CFTR conformational perturbations. The method provides discrete information regarding the solvent accessibility of Lys residues. Only 26 Lys residues out of 92 in CFTR (that contains 1480 amino acids in its five domains) were quantified. Based on very limited number of Lys residues' accessibility changes, the authors suggest novel functional mechanisms of CFTR, conformational changes by CF-causing mutations and the effects of the FDA-approved drugs. However, several concerns are raised due to the limitation of the technique, the selected and incompletely defined experimental conditions, and the over-interpretation some of the data. These weaknesses culminate in the limited novelty of this manuscript regarding CFTR conformational perturbations by activation, mutations, and CFTR modulators.

Major concerns:

1) As stated in the introduction of the manuscript, CFTR undergoes several (~50) posttranslational modifications and can be engaged numerous (>100) protein interactions. Considering that CFTR point mutations, as well as folding correctors provoke both local and long-range global conformational and dynamical changes (extensively documented at the TMD1, NBD1, TMD2 and NBD2 levels [e.g.: PMID 17113596, 15619635, 15619636, 23666117, 33771570], as well as various posttranslational modifications e.g ubiquitination on multiple Lys residues) and perturbations of numerous protein-protein interactions as referenced in the manuscript, based on the available CPP results one cannot conclude that the limited alterations in some Lys accessibility are the results of the channel conformational changes.

2) Although, the F508del mutation indeed causes significant increase and decrease in the K273 and K95 accessibilities, respectively, in relation to the other >60 CFTR Lys residues by CPP, considering the ambiguity of the interpretation of these results (see also point 1), I'm not convinced that these data add any novel information to the previously documented F508del-, modulator-, and activation-induced CFTR conformational perturbations at domain level.

3) The CFTR activity and structural stability have temperature dependence. Since the CPP labeling was performed on ice, the CFTR activity and conformational dynamics would slow down. In addition, the several mutations cause temperature sensitive conformational defects (e.g. F508del), causing conformational rescue at reduced temperature. Thus, the labelling results would not reflect the "native state in vivo". The CPP studies should be performed at the physiological temperature in both corrector rescued and non-rescued mutants.

4) In CPP, bulky dimethyl groups are covalently attached on the side-chain of Lys. Even in 10 min, this modification may profoundly affect the CFTR conformation and function. Could you please include the control experiment and verify that Lys-modifications do not have an effects on CFTR function and global domain conformation.

5) Disappointingly, the authors largely failed to review the large body of data documenting the long-range conformational defect of F508del CFTR (e.g.: Cys-crosslinking, and limited proteolysis with domain specific immunoblotting) and its reversal by corrector molecules (see some examples in point 1 and at the end of the comments).

6) To compare in vivo and in vitro activation of CFTR, HBE cells were exposed to the VX-770 potentiator. VX-770-elicited channel activation requires some level of PKA-dependent CFTR phosphorylation, which was not applied. Thus, under the experimental condition the VX-770-induced channel activation is very limited and suboptimal. In fact, it is surprising that VX-770 alone, in the absence of exogenous phosphorylation, can significantly reduce the localized solvent accessibility of the K273, K370, and K464 residues in WT CFTR. These studies should be performed on phosphorylated CFTR.

7) Based on Fig.2B and Table 1 data, it appears that Lys-residues were over-labelled; the accessibility of four partially accessible Lys residues (based on the cryo-EM) was 96.1-99.7%. The inaccessible residue reached ~62% labelling. Optimization of the labeling time/concentration may be beneficial to reveal additional accessibility differences in CFTR and minimizing the artifact introduced by excessive covalent labelling.

8) References to Figs. were omitted in some of result sections (e.g. I.177-I.186) and abbreviations were inconsistently used (e.g. F508 and F508), exemplifying the lack of proper proof reading of the ms.

9) Please, confirm the Golgi colocalization of the K273A-CFTR mutant by additional biochemical and morphological studies. Another possibility is that the mutant is metastable at the cell surface and accumulates in endosome following its ubiquitination at the PM, endosomes and/or secretory vesicles. This would explain its augmented interaction with the ESCRT machinery by the GO enrichment analysis, a mechanism described for ESCRT-dependent lysosomal targeting of rescued F508del CFTR.

Methodological concerns:

1) In this CPP technique, solvent exposed lys is labelled by CH3 x 2, and solvent excluded lys as well as the N-terminal amine of peptide fragments were labelled by CD3 x 2 as second labelling. The mass of peptides containing the solvent exposed lys shifts by 28.0313 (CH3 x 2) + 36.0757 (CD3 x 2), and that of peptides containing the solvent excluded lys shifts by 36.0757 x 2 (CD3 x 4). Thus, the mass difference between peptides containing the solvent exposed and excluded lys is 8 Da.

In Fig. 1D, the mass spectra of peptides containing K273 were shown. Based on Table 1 and Supplementary Table 2, the peptide containing K273 is VITSEMIENIQSVKAY, and its monoisotopic mass is 1823.93 Da. Since the difference between m/z values in light "inaccessible" peaks is 0.5 (941.509 – 941.0068 = 0.5...), the charge state of this peptide must be z = +2 (therefore, m/z = [M+2H]²⁺). Comparing the m/z values between "light" and "heavy" in Fig 1D, the calculated mass difference is just 4 Da, since (943.016 – 941.0068) x 2 = 4.0184. Could you explain why the difference was not 8 Da?

Moreover, I wonder if authors picked up correct m/z peaks for this peptide. The theoretical monoisotopic m/z value of "light" labelled VITSEMIENIQSVKAY is supposed to be [M + 2H]²⁺ = (1823.93 + 28.0313 + 36.0757 + 2.016)/2 = 945.0265 (indicating almost 4 Da difference from 941.0068). On the other hand, the theoretical mass of "light" labelled VITSEMIENIQSVKAY is 1888.037 Da (= 1823.93 + 28.0313 + 36.0757), and the calculated mass of [M + 2H]²⁺ from 941.0068 is 1879.9976 (= 941.0068 x 2 – 2.016). Likewise, the theoretical mass of "heavy" labelled peptide is 1896.0814, but the calculated mass from [M + 2H]²⁺ = 943.016 is 1884.016. Could you check if authors selected correct mass spectra to determine the surface accessibility?

I suggest tabulating the peptides' masses, m/z values, retention times, charge states, areas in chromatograph used to calculate solvent accessibility and modifications (e.g. how many CH3 and/or CD3 attached) as well as carbamidomethylation of cys, oxidation of met and phosphorylation of Ser and Tyr if peptides contain those) of labelled all peptides.

I am confused the labelling (light "inaccessible" and heavy "accessible") in the Fig 1D. I expected that solvent accessible Lys site is labelled by CH3 (light), and solvent exclude Lys site is labelled by CD3 (heavy). Please clarify the meaning of "inaccessible" and "accessible" in this figure?

2) In Fig 3B, the mass spectra of peptides containing K273 from G551D-CFTR is shown. The charge state must be z = +3 since the m/z difference is 630.6833 – 630.3485 = 0.3348. When I looked at the sequence of the peptide (VITSEMIENIQSVKAY), most likely 2H⁺ attach to the amine groups of lys and the N-terminal, showing highest signal. Indeed, in the Fig. 1D, authors picked up [M+2H]²⁺. Even if [M + 3H]³⁺ was detected, I expect that the signal intensity was much lower than [M+2H]²⁺. Although mass difference between these peaks is 8 Da, since (633.0342 – 630.3485) x 3 = 8.0571, the spectra contain many noise peaks that means low signal intensity. Could you explain why the charge states +3, instead of +2 was used?

3) It is difficult and impossible to find the raw data files related to Figs.1D and 3B. I suggest to : 1) rename the files, indicate the CFTR constructs (WT, F508del, G551D and N1303K) and the condition used (e.g. drugs etc.), and 2) introduce sub-folders for Fig. 1D and 3B in the data base where the raw data files were placed.

4) The CPP was initiated by adding both formaldehyde and sodium cyanoborohydrate to cell, and the labelling time and temperature was 10 min on ice, respectively. I would like to know the labelling rate at this low temperature to label the fully solvent accessible Lys residues of CFTR in cell. Were the labelling conditions of CFTR optimized?

For this, could you monitor the signals of both labelled and unlabeled peptides by only CH3 with time-course, but no second CD3 labelling? Specifically, I would like to see in which time point the MS signals of unlabeled solvent exposed Lys sites are disappeared.

Considering that most of the detected Lys residues in the WT-CFTR are fully accessible after 10 min labeling, shorter covalent labeling would be more informative for reporting CFTR conformational perturbations. Determining the in vivo labeling kinetics should provide an answer for this concern.

5) K95 is in the pore of CFTR in both inwardly and outwardly opened configuration according to the available cryo-EM structures (5UAK and 6MSM). In Fig 1B, the solvent accessibility of K95 of WT-CFTR is 96%. I'm not sure if both formaldehyde and sodiumcyanoborohydrate reach K95 and perform the covalent labelling in the pore. Could you verify that these two molecules are able to reach K95?

7) In p.4, l.136, authors showed that VX-809 changed the solvent accessibility of K1218. This experiment was performed with Δ F508-CFTR. Could you see the same effects on WT-, G551D- and N1303K-CFTR? Could you show if the solvent accessibility change at K1218 by VX-809 is the common effect on other CFTR variants or F508del-CFTR specific effect? If it is the F508del-CFTR specific effect, could you explain why VX-809 change the solvent accessibility of K1218 only for F508del-CFTR, while this corrector has ~not more than 3% rescue efficiency of the mutant processing defects?

8) In Table 1, there are 2 Lys sites (K370 and K377) in the peptide DSLGAINKIQDFLQKQEY. In Fig 1B, only the solvent accessibility of K377 is shown in right panel, on the other hand, only that of K370 is shown in the left panel. Likewise, only the surface accessibility of K370 was determined in Fig.2D. Could you explain how did you quantify the solvent accessibility if peptides have multiple Lys sites? If you can separately determine the solvent accessibility of K370 and K377, both Lys accessibilities should be shown in Fig.1B right and left panels?

9) Fig. 2C shows the reduced solvent accessibility of K273 of WT-CFTR in the presence of VX-770. Authors claimed the coupling of ICL2 to NBD2 upon activation with VX-770. Can similar results be obtained in G551D-CFTR?

Minor concerns:

1) In introduction (p.2, l.62), authors stated that "...several of which are located in the small disordered region, the RI element, is deleted in recombinantly expressed CFTR used for the Cryo-EM studies.", but many of available cryo-EM structural determinations were carried out on CFTR channels that contain the RI region, although the disordered region is not visible. This statement could be misleading.

2) Ref 9 was published in a peer-reviewed journal, on the other hand, ref 13 was published in bioRxiv.

3) In p.3, l. 98, "Twenty-six lysine sites in CFTR were quantified in 3 different biological replicates..... (Fig. 1B, 1C; Table 1). However, the figure legend indicates: Wt CFTR n=4; Δ F508 CFTR n=4 (biological replicates), p11, l.481. Moreover, in Fig 1B, many data points in each lysine site are shown. Could you explain why there are many data points although biological replicates were 3 or 4? Please explain what the error bars are.

4) In Fig 1B, 2B and 2D, F508 should be F508del.

5) In p.4, l.151, PDB 5UAR is the cryo-EM structure of zebrafish CFTR with dephosphorylated, ATP-free states. Please use the right PDB file to determine the solvent accessibility of K273 in human CFTR. Also, ref 22 is the literature of zebrafish CFTR dimerized forms. Please cite the literature of human CFTR dimerized forms.

6) l.318 Structural differences not only in the NBD2 but all structured domains of the F508del-CFTR have been documented in relation to that of the WT-CFTR by several groups, please include appropriate references.

7) Could you please clarify the meaning(s) of following sentence.

l. 327: Point mutation of K1218 to K1218R has been shown to increase the amount of CFTR, stabilize it at the plasma membrane and reduce lysosomal degradation of CFTR, while its deletion improves folding efficiency of Δ F508 CFTR.

Reviewer #2

(Remarks to the Author)

Summary:

Pankow and colleagues present an application of their Covalent Protein Painting (CCP) method (Bamberger et al., 2021 J. Proteome Research), to the challenge of understanding the conformational landscape of Cystic Fibrosis Transmembrane Regulator (CFTR). As a protein of therapeutic interest, CFTR has a range of well-characterised conformational variants. The authors exploit the abundance of structural data, model cell systems, and chemical/physical modifiers available for CFTR to contextualise their CCP findings and reveal novel conformations of CFTR abundant in the cell. While this is a nice showcase of the methodological capabilities, the importance of these findings for a broad readership is unclear. Unfortunately, critical weaknesses in, and the description provided for, several data visualisations presented throughout the manuscript prevent these results from being adequately examined and interpreted. The authors also do not provide robust experimental controls for the tagged CFTR variant work and essential methodological information is missing which would

preclude replication of the study as a whole. Together, I believe these issues render the work as presented unsuitable for publication in Nature Communications.

Major concerns:

1. Data presented in Figures 1B, 2B, 2D, 3A, S1A, S1B: Each of these panels is a variation of scatterplot visualizing the accessibility of specific residues under a given condition. While the authors' effort to include individual data points is acknowledged, the plots are extremely challenging to interpret. The data summary elements (mean, SD/SEM?) often cannot be distinguished from the replicate data points, and the results referred to in the text as having significant differences are not annotated. The overlap in different treatments (especially Figure 2D) makes differences between the treatments exceptionally difficult to evaluate. In addition, the link between the number of replicates specified in text, in the figure legend and individual data points shown on the graph is unclear. For example, in Figure 1B, $n=3$ in the main text, $n=4$ in the figure legend, and the actual plot shows some lysines with upwards of 10 individual markers in a single experimental condition. One assumes these might come from quantification of different peptides containing the lysine of interest however this is not described in the legend or methods. In addition, the summary elements are not described in the legend making it unclear if the error bars represent SEM/SD or something else, and therefore it is not possible to rationalise why some error bars are unidirectional while others are present both above and below.
2. Experimental details concerning the cell lines, and the rationale behind the use of each of the different cell lines throughout the course of the study, are lacking. How are the CFTR variants expressed in the CFBE/HBE cells? Why is the K273 mutation then profiled via transient overexpression in a non-specialist cell type (HEK293)? Different cell models likely impact the conformation and available interactome, do the authors expect their results to be generalisable to the specialist cell types?
3. Figure 1B: There appear to be substantially fewer lysines quantified in the label swap experiment - why is this? Does this point toward an inherent bias in the method whereby one version is favoured for best coverage? Similarly, some lysines appear to be quantified only in one of the cell lines, were these peptides not identified in any of the replicates for that cell line? How does this affect the conclusions made about these residues with respect to solvent accessibility?
4. Figure 2: The legend for panel A only discusses one drug but the heatmap appears to present several combinations. There also appears to be the cell lines alone (WT, d508) but it is not clear in the legend or figure whether the treatments are being conducted on the WT or mutant background. In addition, the colorbar labels are not adequately sized or specific (what 'value' is shown?). Similarly, panel D includes the following legend descriptors: wt, wt_VX770, F508, Trikafta, VX445-VX661, F508_28 celsius. This does not make sufficiently clear which conditions are applied to which cell lines. It is not clear to which comparisons the annotated p-values refer to, nor the other annotated summary measures.
5. The immunoprecipitation experiment (Fig. 4C, D) uses GFP as the control protein against which the interactome of the K273 is determined, however, this does not discriminate native interactions maintained by the variant from those specific to the conformation modelled by the K273 mutation. A WT comparison is required to understand the specific interactions associated with the novel conformation explored in this study.
6. Figure 4: The network interaction map provided in Figure 4C is illegible at standard zoom and not adequately described in the legend (e.g. what do the directional arrows represent?). Similarly, how were the proteins presented in Figure 4D selected from the list of interactors – what does 'specifically enriched' mean? Were they the most significantly enriched? And were all the proteins presented identified in the interactome? This appears to be a protein-protein interaction map, what database was used? Was any filtering for confidence in those interactions completed? These details are missing from the methods. The legend for this figure is similarly lacking in important details, for example how the protein nodes are colored and sized, and by what mechanism the proteins presented were arranged/clustered. Together these issues prevent accurate interpretation of the data.

Minor comments:

1. Data tables as provided in the supplementary material are in a pdf-embedded format. Please provide a format more suited to tabular data e.g. csv to improve the openness and reusability of the dataset.
2. Discussion of the importance or relevance of these findings to the broader audience, or even to the CF field where one imagines the novel conformation is of most interest, is limited.
3. Figure S2 contains no evidence of magenta staining at all. Is general diffuse staining expected (e.g. for LC3, as observed in HeLa)? And if not, a positive control is required to demonstrate successful staining and prove the absence of reactivity in the cell lines of interest. In addition, the legend indicates a comparison with WT cells but the panels presented are both labelled K273.
4. Line 212: "Appears to be Golgi-related structures". This statement is not supported by immunocytochemistry, only the absence of autophagy markers. Excellent markers exist for the Golgi compartments.
5. Method descriptions –brief method outlines should be preferred over references to aid reproduction of the method as applied in this specific study (line 362, Line 367, Line 375, line 380, line 427).

6. Readability – extended sentences and grammatical structure should be revised, for example, lines 40-44, lines 128-136.

7. Line 120 – 121: Comparing solvent accessibility with solvent inaccessibility makes interpreting this result more challenging than necessary. Please compare apples with apples, e.g. “Solvent-accessible in only 60% of wt CFTR molecules, but to greater than 95% in Δ F508 CFTR”.

8. Descriptions for each of the cell lines used should be provided in text at first mention, followed by the abbreviation to be used throughout the remainder of the manuscript. Line 89 human bronchial epithelial cells expressing wt CFTR (HBE16o-) abbreviated later to HBE, Line 107 – isogenic CFBE41o- cells expressing Δ F508 CFTR abbreviated in methods to CFBE. Similarly for other abbreviations used throughout the manuscript (e.g. GO is not defined).

Reviewer #3

(Remarks to the Author)

Reviewer #4

(Remarks to the Author)

This manuscript describes the application of a protein foot-printing approach using lysine labelling to characterize CFTR conformational changes associated with various Cystic Fibrosis disease causing mutations in CFTR as well as with various drug treatments. The so-called covalent protein painting (or CPP) approach used in this work has been previously described and used in other applications by the authors. One attractive and unique feature of CPP is that it can be used to study proteins in vivo, which is done in the current manuscript. Another important feature of covalent labelling strategies like CPP is that they can capture “dynamic” information about protein structure that is missed using other structural biology techniques like x-ray crystallography and Cryo-EM. In fact, this is demonstrated/highlighted in the current work where a unique CFTR conformation involving the ICL2 and NBD2 domains of CFTR is detected. As part of the work described in the current manuscript the authors investigate the disease relevance of this conformation, which can be specifically probed by measuring K273 protection in CFTR. Their studies on various disease mutants and drug treatments reveal that the newly discovered solvent-exposed K273 conformation of CFTR is an inactive conformation that is also observed in deltaF508 and N1303K CFTR mutants. The authors' CPP data also provided for a better understanding of VX-809 drug mode-of-action.

Overall, the work is well-down and well-presented, with a few exceptions noted below. The findings are not only significant from a technique development perspective (e.g., the demonstration of CPP unique ability to detect CFTR conformations not previously captured using other more conventional structural biology methods), but it is also significant from a CFTR disease biology perspective (e.g., the new details that are learned about VX-809's mode of actions, and the newly discovered presence of a biologically relevant protein conformation in CFTR). Thus the work stands to be of great interest to the general readership Nature Communications, where it would be appropriately published (after minor revision-see below).

Other more specific comments that need to be addressed before publication:

Line 31: should be “identifies”

Line 90: not clear why “(CPP)” is placed at the end of the sentence? Maybe it should read, “...according to the CPP workflow in Fig 1A.”

Line 124: Should include references to the Cryo-EM structures.

Line 200: delete “well as”

Lines 310 and 320: The last 2 paragraphs both begin with “Finally”. Probably just the last paragraph should begin with “finally.”

Refs 9 and 11 seem to be to the same paper. Maybe just used the peer-review publication in JPR (ie., ref 9).

In the Figure panels showing the CPP solvent accessibility data, it is not clear what the actual points represent. Seems like these should be the data from the biological replicates? But this does not seem to be the case because the numbers don't match up. This needs to be better explained in the legends.

Figure 2 is confusing especially since it contains 4 panels but the legend only describes 3. And the existing description of the first three panels is confusing (almost to the point that it doesn't seem to be describing the existing panels). This needs to be fixed/clarified.

Version 1:

Reviewer comments:

Reviewer #1

(Remarks to the Author)

Please, see attached file.

Reviewer #2

(Remarks to the Author)

Pankow and colleagues present an application of their Covalent Protein Painting (CCP) method (Bamberger et al., 2021 J. Proteome Research), to the challenge of understanding the conformational landscape of Cystic Fibrosis Transmembrane Regulator (CFTR). As a protein of therapeutic interest, CFTR has a range of well-characterised conformational variants. The authors

exploit the abundance of structural data, model cell systems, and chemical/physical modifiers available for CFTR to contextualise their CCP findings and reveal novel conformations of CFTR abundant in the cell. While this is a nice showcase of the methodological capabilities, the importance of these findings for a broad readership is unclear. Efforts to add detail to the methods and figure legends have improved the manuscript substantially. However, several aspects of the previous review are yet to be addressed and clarification is required to confirm the data supports the manuscripts conclusions.

1. Major Concern #3 remains unresolved. The authors state "The label swap experiment primarily ensures labelling consistency." (rebuttal), and "A label swap experiment confirmed the detected differences between wt and Δ F508 CFTR" (Line 116). However, some residues disagree in solvent accessibility by up to 10% between the two orientations, and there are differences in whether the residue was quantified in only one or both of the cell lines. Importantly, this difference appears in one of the residues highlighted by the authors (K95), where the Light-heavy F508 is ~76% (compared to WT 97%) and the Heavy-light F508 is ~86%, while the Heavy-light WT is not quantified. This example highlights that not all peptides are consistent, and reinforces the importance of two questions not addressed by the authors in this revision: *Some lysines appear to be quantified only in one of the cell lines, were these peptides not identified?* and *How does this affect the conclusions made about these residues with respect to solvent accessibility?* To elaborate, the authors state that "Both sites were less accessible in F508 CFTR" (line 123), however, it is unclear how the effect of the F508 mutation can be described in this manner without the WT measurement to compare to.

2. "We checked if any error bars were unintentionally set to unidirectional, but this was not the case and we believe that some error bars were plotted erroneously as unidirectional by the software due to plotting space limitations. We have now used a newer version of this software." (rebuttal). However, Figure 1B still contains unidirectional error bars (Light-heavy) and lacks error bars entirely (Heavy-light), and Supplemental Figure 1 contains unidirectional error bars.

3. "We have now clarified this in both the main text and the figure legends... Additionally, we have enhanced the figures by clearly annotating statistical significance and error bars (mean \pm SEM)." (rebuttal). Figure 3A contains error bars that are not described in the legend.

4. The decision to use different cell lines is now well justified (provided in response to Major Concern #2). Please include this information in the manuscript as it provides important context required to interpret the presented results for readers from a general audience.

5. The labelling for Figure 2A is still unclear. The legend states "Heatmap displaying solvent accessibility changes of Δ F508 CFTR upon treatment of CFBE41o- cells with Trikafta (24 h), the two corrector compounds VX-661 (5 μ M, 24h) and VX-445 (3 μ M, 24h), corrector VX-809 (5 μ M, 36h), at permissive temperature of 28 °C (24h), or in DMSO solvent controls and of wt CFTR." (line 513). However, the labels on the heatmap and scatterplot contain "F508". How is this different to the DMSO solvent control (labelled DMSO)?

6. Figure 2D remains uninterpretable owing to overlap in different treatments. While the means are now provided, the authors should consider alternative visualisation options e.g. individual scatterplots for each lysine residue where the different treatments can then be visualised as separate columns.

7. Figure 4C: Text size for the node labels remain unreadable. More importantly, components of the visualisation are still not adequately described. The authors state "The arrows represent directionality between categories as far as we are aware." (rebuttal). What is directionality i.e. what does the direction of the arrows represent when linking two nodes? They appear to link hierarchical elements of the ontology, but this is not described. This needs to be clarified.

Additional concerns:

8. The inclusion of biological information is commendable. However, it raises a question regarding the validity of statistical comparisons. The actual n for each residue is a combination of biological replicates and the number of distinct peptides containing that residue, which can vary significantly (some have 1 data point, some > 10). Since increased n can correlate with a higher likelihood of detecting significant results (Shreffler & Huecker. "Type I and Type II Errors and Statistical Power." StatPearls, 2024), could the authors clarify how they address the potential confounding effects of combining replicate and peptides to artificially increase n? This method also obscures variability between biological replicates/conditions, potentially affecting the interpretation of visualisations.

9. Several observed differences (e.g. Figure 1B) appear to be based on data from a single peptide within a single replicate (one data point). Please elaborate on the reliability of these differences, particularly given they are detected in only 1 out of 4 biological replicates - how does this impact confidence in the findings for these data?

10. Supplemental Figure 4 is a table (not a figure)

Reviewer #3

(Remarks to the Author)

Reviewer #4

(Remarks to the Author)

This reviewer has no further concerns.

Version 2:

Reviewer comments:

Reviewer #2

(Remarks to the Author)

The authors' efforts to incorporate the feedback from the previous rounds of review are acknowledged and appreciated. Please find below some final comments relating to the presentation of the data in the updated manuscript:

1. Figure 2D: The y-axis label is overlapping and needs to be adjusted.
2. Supplemental Figure 3: Scale bars should be included for clarity.
3. Supplementary Tables 1 & 2: These tables are still provided in PDF format, which was previously raised in the first round of review as Minor Concern #1. Providing the data tables in a format more suited to tabular data, such as CSV, would improve the openness and reusability of the dataset.
4. Figure 4C (Major Concern #7): The labels in this figure remain unreadable. Additionally, while the zip folder contains XML files for the relevant Cytoscape session, the .cys file is missing, making it unclear how to view this network for readers without expertise. More importantly, main text figures should be readable and interpretable without the need to consult an external program. Please either include this panel at a size that makes the labels legible or consider removing it from the main text.
5. Peptide Quantification (Additional Concern #9): While your efforts to confirm the differences for these peptides are commendable, the limitations of the technique that may underlie the failure to quantify these peptides more robustly need to be addressed. The issue that these peptides were observed in a single biological replicate should not be discounted. Including a discussion on the reliability of such measurements, along with the expected differences in the label-swap experiment, would help alert readers to the limitations of the conclusions drawn from these data.

Best of luck with the remainder of the publication process.

Reviewer #5

(Remarks to the Author)

Comments on author rebuttal on reviewer #1's comments.

Response to comment 1

A crucial point is raised by the reviewer on the actual conformational changes that are monitored by the authors. This highly relevant question relates to (mostly co-translational) events that occur early during folding and (mostly post-translational) interactions between domains leading to a functional conformation. The manuscript does not contain experimental data to distinguish between these modalities. The authors' reply is basically a textbook summary of all intracellular itineraries a newly synthesized secretory protein can undergo from its synthesis on the ribosome to degradation by any of the proteolytic machineries of the cell. The field has advanced much farther though than both manuscript and rebuttal acknowledge. Not only was reviewer's comment not addressed, the response suggests that these authors lack textbook expertise on the biogenesis of proteins in the secretory pathway, and of published knowledge on CFTR biogenesis and transport by a wealth of laboratories including those of Riordan, Cyr, Gentsch, Amaral, Braakman, Clarke, Lukacs, Ph. Thomas.

Response to comment 2

Reviewer is correct that N1303K shows variable responses, both clinically and in vitro, probably because of the variable maturation (C band) shown in different publications, also defined as variable 'residual activity'. The response is not minimal, as replied by authors, so this should be corrected in the manuscript. N1303K is FDA-approved for Trikafta.

Response to comment 3

Reviewer asks to provide evidence in support of a claimed novel opening mechanism. It is legitimate to ask for independent information on CFTR phosphorylation status under these conditions, since it is important for the interpretation of the results. The authors refer to a lengthy explanation given under comment 8, but this does not bear relevance to the point reviewer

addresses. Neither manuscript text nor response are clear, as there is inconsistent use of terms like 'CFTR activation' when VX-770 is added to the cells, 'active CFTR', and the open conformation. This does not help the reader.

Response to comments 4 and 8

The reviewer questions the significance of increased accessibility of K273 and F508del citing knowledge described in several previous publications, which is a fair and crucial point. Authors reply that these studies were done on recombinant CFTR or in vitro, which is incorrect and does not do justice to work that has been done in live cells for decades by other laboratories. Also incorrect is the claim in the rebuttal that 'this is the first in vivo study that can point out structural differences on a residue level. The Braakman-Van der Sluijs lab described an in-cell folding assay for CFTR and used it on various CF-causing missense mutants [Im et al. Cell Mol Life Sci. 2023. Jan 7;80(1):33. doi: 10.1007/s00018-022-04671-x. PMID: 36609925; PMID: 36609925; PMID: 33771570; PMID: 30659068; PMID: 36499495]. They showed amongst others that CFTR-F508del cannot fold NBD1 and as a consequence fails to assemble the domains, and that the Vertex correctors rescue domain assembly and not NBD1 folding. ICL2 becomes protease-protected only during domain assembly, whereas in newly synthesized CFTR ICL2 is accessible to protease in all CFTR variants studied. What this means is that solvent accessibility or inaccessibility of K273 may not only, or perhaps may not be at all relevant for activity of the channel, but rather reports on early and late steps of CFTR (variant) folding! The focus of the authors on structural re-arrangements involved or required for activation (without additional supporting experimental evidence) therefore presents a major conceptual shortcoming of the manuscript. The K273A data to some extent are consistent with in-cell publications, but the authors mix discussions on activation, open-closed channels, VX-770 effects, and biogenesis such as domain folding and domain assembly.

Response to comment 5

Reviewer is concerned that the structural integrity of CFTR variants might be (differentially) affected by the labeling procedure. An important shortcoming of the manuscript is the lack of independent experimental validation of reported findings. Authors reply that the mass-spec method has been validated on '>10,000 proteins that agree with PDB data'. Three points deserve consideration, the first of which is that PDB mostly reports on stable forms of a protein, which are not transient or intermediate (folding) conformations. Secondly, only for a limited number of proteins has a folding pathway been elucidated in cells, these should be used to benchmark the method. Thirdly, how many of these >10,000 proteins are complex multidomain-multispanmembrane proteins?

Response to comment 6

Reviewer has a justified point to be concerned about the temperature at which the labeling experiments are performed. The authors reply with theoretical considerations that do not take into account the complexity of a cell. For instance, temperature-dependent alterations of membranes in a cell cannot be taken to be realistically the same as in 'isolated lipid particles'. For instance, a 15°C incubation blocks transport from ER to Golgi, whereas 20°C blocks transport at the level of the TGN. Membrane transitions are known to occur at 18 and 21 °C in cells, which leads to stiffening of the membranes. The authors show repeatedly their naivete on cellular processes.

Response to comment 9

Reviewer states that resolution of images in S2e are poor. In fact, resolution is poor in all images and cannot tell more than that distribution of wt and K273A is different. Conclusions on colocalization or lack thereof with ER and Golgi markers is not justified because the signals of the markers are quite diffuse already and do not lend themselves for quantitative assessment. Moreover, the structures for K273A may well be aggregates rather than post-ER compartments. The precise intracellular localization of the CFTR variants remains to be established; only cell surface is relatively convincing. Although the authors seem to be aware of this, they nevertheless attempt to draw conclusions by seeking validation from interactome studies, but this is problematic. How does a claimed Golgi localization of K273A square with an interactomes enriched in proteins controlling membrane-transport pathways between endocytic organelles? Determining location of a protein through its interactome is too inaccurate, especially in steady-state experiments. Independent experimental information is needed for rigorous conclusions.

REVIEWER COMMENTS

Response to Reviewer 1. We appreciate the reviewer's insightful comments and have provided clarifications and addressed concerns in detail under each point raised.

Reviewer #1 (Remarks to the Author):

In this study, the authors interrogate the in vivo conformational changes of four CFTR variants (wild type [WT], Δ F508, G551D and N1303K) by utilizing the previously developed and published Covalent Protein Printing (CPP) method. The CPP is based on the determining the extent of covalent modification of exposed epsilon amines of Lys residues by MS quantification that was applied here for probing CFTR conformational perturbations. The method provides discrete information regarding the solvent accessibility of Lys residues. Only 26 Lys residues out of 92 in CFTR (that contains 1480 amino acids in its five domains) were quantified. Based on very limited number of Lys residues' accessibility changes, the authors suggest novel functional mechanisms of CFTR, conformational changes by CF-causing mutations and the effects of the FDA-approved drugs. However, several concerns are raised due to the limitation of the technique, the selected and incompletely defined experimental conditions, and the over-interpretation some of the data. These weaknesses culminate in the limited novelty of this manuscript regarding CFTR conformational perturbations by activation, mutations, and CFTR modulators.

Response: We would first like to reference our previously published papers on Covalent Protein Painting (CPP) and include references to dimethyl labeling for peptides to clarify any confusion about the labeling mechanism and why certain features are essential. Dimethyl labeling is a chemical labeling method that allows for **complete** labeling of the e-amine groups of lysines with exactly two dimethyl groups and can be achieved in a very fast time (less than 15 s). This process is well-documented, such as in Boersema et al. (2009) and in Bamberger et al., 2021. Additionally, publications on the analysis of mass spectrometric data, including Xu et al. (2015) and Yates et al. (1995), should help clarify the data analysis methods employed. Key references include:

- Bamberger C, Pankow S, Martínez-Bartolomé S, Ma M, Diedrich J, Rissman RA, Yates JR 3rd. Protein Footprinting via Covalent Protein Painting Reveals Structural Changes of the Proteome in Alzheimer's Disease. *J Proteome Res.* 2021 May 7;20(5):2762-2771. doi: 10.1021/acs.jproteome.0c00912.
- Boersema PJ, Raijmakers R, Lemeer S, Mohammed S, Heck AJ. Multiplex peptide stable isotope dimethyl labeling for quantitative proteomics. *Nat Protoc.* 2009;4(4):484-94. doi: 10.1038/nprot.2009.21.
- Bamberger C, Diedrich J, Martínez-Bartholomé S, Yates JR 3rd. Cancer Conformational Landscape Shapes Tumorigenesis. *J Proteome Res.* 2022 Apr 1;21(4):1017-1028. doi: 10.1021/acs.jproteome.1c00906
- Son A, Pankow S, Bamberger TC, Yates JR 3rd. Quantitative structural proteomics in living cells by covalent protein painting. *Methods Enzymol.* 2023;679:33-63. doi: 10.1016/bs.mie.2022.08.046
- Xu *et al.*, ProLuCID: An improved SEQUEST-like algorithm with enhanced sensitivity and specificity. *JPR* (2015).
- Yates *et al.*, Method to correlate tandem mass spectra of modified peptides to amino acid sequences in the protein database. *Anal. Chem* (1995).

Major concerns:

1) As stated in the introduction of the manuscript, CFTR undergoes several (~50) posttranslational

modifications and can be engaged numerous (>100) protein interactions. Considering that CFTR point mutations, as well as folding correctors provoke both local and long-range global conformational and dynamical changes (extensively documented at the TMD1, NBD1, TMD2 and NBD2 levels [e.g.: PMID 17113596, 15619635, 15619636, 23666117, 33771570], as well as various posttranslational modifications e.g ubiquitination on multiple Lys residues) and perturbations of numerous protein-protein interactions as referenced in the manuscript, based on the available CPP results one cannot conclude that the limited alterations in some Lys accessibility are the results of the channel conformational changes.

2) Although, the F508del mutation indeed causes significant increase and decrease in the K273 and K95 accessibilities, respectively, in relation to the other >60 CFTR Lys residues by CPP, considering the ambiguity of the interpretation of these results (see also point 1), I'm not convinced that these data add any novel information to the previously documented F508del-, modulator-, and activation-induced CFTR conformational perturbations at domain level.

Response to comments 1 and 2. Lysines modified by ubiquitination or other PTMs are not labeled by the CPP technique and thus would not appear in our data. Only labeled lysines contribute to the solvent accessibility ratio. Thus, any changes in lysine accessibility ratios reported here are due to channel conformational changes and cannot be attributed to PTMs. Furthermore, previous studies including our own (Lee et al., 2014; Pankow et al, 2019) indicate that the proportion of CFTR modified by ubiquitination is typically less than 1% and would not contribute to any larger changes reported in the manuscript. The low abundance of PTMs is a well known fact in the proteomics field and PTMs typically require specialized enrichment procedures to become visible to the mass spectrometer (see Zhao and Jensen, 2009) Furthermore, Lys276, which shows the largest conformational changes in response to perturbation, has never been identified as post-translationally modified in any previously published dataset or this dataset. While we acknowledge the complexity of interpreting accessibility data, the results reported here are not ambiguous at all and reveal novel insights into F508del-related CFTR conformational dynamics, particularly concerning the accessibility of K273. We believe that our observations contribute valuable information to the growing body of research on CFTR's structural changes induced by mutations, modulators, and channel activation.

3) The CFTR activity and structural stability have temperature dependence. Since the CPP labeling was performed on ice, the CFTR activity and conformational dynamics would slow down. In addition, the several mutations cause temperature sensitive conformational defects (e.g. F508del), causing conformational rescue at reduced temperature. Thus, the labelling results would not reflect the "native state in vivo". The CPP studies should be performed at the physiological temperature in both corrector rescued and non-rescued mutants.

Response. CFTR's temperature-dependent conformational changes typically occur over several hours (e.g., glycosylation changes are observed over 24–48 hours, see Denning et al, Nature, 1972). The CPP labeling is complete after 15 seconds (see Bamberger et al, 2021 JPR) and thus the temperature at which labeling occurred is unlikely to significantly impact CFTR's conformation during the experiment. It is like a "snapshot of the conformational states of CFTR". It is also unlikely that the temperature in the cell culture dish is severely reduced within 15 seconds of adding the label.

4) In CPP, bulky dimethyl groups are covalently attached on the side-chain of Lys. Even in 10 min, this modification may profoundly affect the CFTR conformation and function. Could you

please include the control experiment and verify that Lys-modifications do not have an effects on CFTR function and global domain conformation.

Response: We respectfully disagree with the notion that dimethyl groups are bulky. Dimethyl groups are very small labels that are covalently attached to the epsilon amine of the lysine and are very unlikely to affect CFTR conformation by themselves. A proof-of-principle control experiment showing that labeling itself did not influence protein conformation was published previously in Bamberger et al., 2021. No functional measurements or experiments are performed after the label was added. After performing the labeling and quenching the labeling reaction, the native protein state is “frozen” in the cells, the cells will not be viable and physiological measurements can no longer be performed. We hope that the confusion about the CPP technique and dimethyl labeling in proteomics is clarified by the references included at the beginning.

5) Disappointingly, the authors largely failed to review the large body of data documenting the long-range conformational defect of F508del CFTR (e.g.: Cys-crosslinking, and limited proteolysis with domain specific immunoblotting) and its reversal by corrector molecules (see some examples in point 1 and at the end of the comments).

Response: We apologize for having to omit most of the fantastic literature on the long-range conformational changes in $\Delta F508$ CFTR observed by Cys-crosslinking and limited trypsin digest due to space limitations. We have added a few select citations suggested by the reviewer. We wish we could include and discuss more of these data, perhaps this could be included in an additional manuscript.

6) To compare in vivo and in vitro activation of CFTR, HBE cells were exposed to the VX-770 potentiator. VX-770-elicited channel activation requires some level of PKA-dependent CFTR phosphorylation, which was not applied. Thus, under the experimental condition the VX-770-induced channel activation is very limited and suboptimal. In fact, it is surprising that VX-770 alone, in the absence of exogenous phosphorylation, can significantly reduce the localized solvent accessibility of the K273, K370, and K464 residues in WT CFTR. These studies should be performed on phosphorylated CFTR.

Response: The VX-770 experiments were performed in living cells in culture similar to the situation in patients, and not with isolated CFTR in vitro. We apologize, if this was not abundantly clear from the description. VX-770 has been shown to be effective in cultured cells (Van Goor et al., 2009) and phosphorylation occurs in cells, which is why VX-770 is effective under these conditions.

7) Based on Fig.2B and Table 1 data, it appears that Lys-residues were over-labelled; the accessibility of four partially accessible Lys residues (based on the cryo-EM) was 96.1-99.7%. The inaccessible residue reached ~62% labelling. Optimization of the labeling time/concentration may be beneficial to reveal additional accessibility differences in CFTR and minimizing the artifact introduced by excessive covalent labelling.

Response: The term “over labeling” may reflect a misunderstanding of the technique. In CPP, exactly two dimethyl groups are attached to each lysine epsilon group. There is no concept of “over labeling” in this context. The numbers given in the manuscript do not reflect the percentage of labeling, but reflect the ratio between solvent accessibility to solvent exclusion. Two methods papers (cited in the relevant methods section and throughout the manuscript (Bamberger et al,

2021 and Son et al., 2023)) explain the CPP technique in detail and numerous papers explain the dimethyl labeling per se in detail (see Boersema et al, 2009 for example). In addition, the e-amine group of most lysines is fully solvent accessible consistent with the presented data (see for example, Hermanson, G.T. *Bioconjugate Techniques*; Academic Press: Boston, MA, USA, 2013).

8) References to Figs. were omitted in some of result sections (e.g. I.177-I.186) and abbreviations were inconsistently used (e.g. F508 and Δ F508), exemplifying the lack of proper proof reading of the ms.

Response: We sincerely apologize for any oversight. We have customarily added the references to Figures at the end of the description of an experimental result, rather than at the beginning, but are happy to change this if needed. The Δ symbol was used consistently throughout the manuscript, but some browsers and pdf readers may not properly display it. We have now checked that the latest version of Acrobat is used to ensure that most readers will correctly display the symbol.

9) Please, confirm the Golgi colocalization of the K273A-CFTR mutant by additional biochemical and morphological studies. Another possibility is that the mutant is metastable at the cell surface and accumulates in endosome following its ubiquitination at the PM, endosomes and/or secretory vesicles. This would explain its augmented interaction with the ESCRT machinery by the GO enrichment analysis, a mechanism described for ESCRT-dependent lysosomal targeting of rescued F508del CFTR.

Response: We would like to thank the reviewer for raising this excellent point. We agree with the reviewer that the K273A-CFTR mutant could be metastable at the cell surface and that enhanced interaction with the ESCRT machinery might result from endosomal accumulation. Therefore, we performed additional immunofluorescence studies using a Golgi/endosome live stain (Golgi-red detection probe, BioLegend), which shows partial overlap with the K273A CFTR mutant (Supplemental Figure 2). We also included additional pictures showing significant overlap of the GFP signal of K273A CFTR with the ER using the ER Cytopainter kit, which selectively stains the ER and partially the Golgi apparatus (Supplemental Figure 2). Additional staining with lysosomal staining reagent (Cytopainter) showed no overlap between lysosomes and the K273A-CFTR mutant, suggesting that the majority of the K273A CFTR mutant is retained in the ER/Golgi.

Methodological concerns:

1) In this CPP technique, solvent exposed lys is labelled by $\text{CH}_3 \times 2$, and solvent excluded lys as well as the N-terminal amine of peptide fragments were labelled by $\text{CD}_3 \times 2$ as second labelling. The mass of peptides containing the solvent exposed lys shifts by $28.0313 (\text{CH}_3 \times 2) + 36.0757 (\text{CD}_3 \times 2)$, and that of peptides containing the solvent excluded lys shifts by $36.0757 \times 2 (\text{CD}_3 \times 4)$. Thus, the mass difference between peptides containing the solvent exposed and excluded lys is 8 Da .

In Fig. 1D, the mass spectra of peptides containing K273 were shown. Based on Table 1 and Supplementary Table 2, the peptide containing K273 is VITSEMIENIQSVKAY, and its monoisotopic mass is 1823.93 Da. Since the difference between m/z values in light “inaccessible” peaks is 0.5 ($941.509 - 941.0068 = 0.5\dots$), the charge state of this peptide must be $z = +2$ (therefore, $m/z = [M+2H]^{2+}$). Comparing the m/z values between “light” and “heavy” in Fig 1D,

the calculated mass difference is just 4 Da, since $(943.016 - 941.0068) \times 2 = 4.0184$. Could you explain why the difference was not 8 Da?

Moreover, I wonder if authors picked up correct m/z peaks for this peptide. The theoretical monoisotopic m/z value of “light” labelled VITSEMIENIQSVKAY is supposed to be $[M + 2H]^{2+} = (1823.93 + 28.0313 + 36.0757 + 2.016)/2 = 945.0265$ (indicating almost 4 Da difference from 941.0068). On the other hand, the theoretical mass of “light” labelled VITSEMIENIQSVKAY is 1888.037 Da ($= 1823.93 + 28.0313 + 36.0757$), and the calculated mass of $[M + 2H]^{2+}$ from 941.0068 is 1879.9976 ($= 941.0068 \times 2 - 2.016$). Likewise, the theoretical mass of “heavy” labelled peptide is 1896.0814, but the calculated mass from $[M + 2H]^{2+} = 943.016$ is 1884.016. Could you check if authors selected correct mass spectra to determine the surface accessibility? I suggest tabulating the peptides’ masses, m/z values, retention times, charge states, areas in chromatograph used to calculate solvent accessibility and modifications (e.g. how many CH₃ and/or CD₃ attached) as well as carbamidomethylation of cys, oxidation of met and phosphorylation of Ser and Tyr if peptides contain those) of labelled all peptides. I am confused the labelling (light “inaccessible” and heavy “accessible”) in the Fig 1D. I expected that solvent accessible Lys site is labelled by CH₃ (light), and solvent exclude Lys site is labelled by CD₃ (heavy). Please clarify the meaning of “inaccessible” and “accessible” in this figure?

Response: An excellent resource on how to correctly calculate the mass differences of dimethyl labeled peptides is the Nature protocols paper from Boersema et al., 2009. For our specific case, the theoretical mass difference between the two labels is 8.04 Da as the N-termini of the light and heavy labeled peptides are both modified by ¹³CD₃ groups. Therefore, in a charge 2 peptide the observed mass difference is 4.02 Da as correctly displayed. The example spectra shown in Figure 1D are from the “label swap” experiment described in the manuscript and as clearly indicated in the figure itself. In this case, in vivo solvent accessible lysines were labeled “heavy” and solvent excluded lysines as well as N-termini were labeled “light”. Of note, the mass difference of course remains the same. The calculated monoisotopic mass for the heavy labeled peptide is 1881.0038 Da and the measured $M+H$ was 1881.0079 Da. The observed mass difference of the identified peptides to the calculated mass difference is less than 1.6 ppm, which is minimal. Of course, peptides and areas are not handpicked among the 10,000s of spectra either. Peptides were identified by the very well established search engine ProLuCID (Xu *et al.*, ProLuCID: An improved SEQUEST-like algorithm with enhanced sensitivity and specificity. Journal of Proteomics 2015. ProLuCID search parameters were set to allow minimal mass differences of less than 5 ppm. Spectra and raw files containing these information have been deposited into MASSIVE as it would be very impractical to report these measurements for the tens of thousands of measured peptides in a table.

2) In Fig 3B, the mass spectra of peptides containing K273 from G551D-CFTR is shown. The charge state must be $z = +3$ since the m/z difference is $630.6833 - 630.3485 = 0.3348$. When I looked at the sequence of the peptide (VITSEMIENIQSVKAY), most likely 2H⁺ attach to the amine groups of lys and the N-terminal, showing highest signal. Indeed, in the Fig. 1D, authors picked up $[M+2H]^{2+}$. Even if $[M + 3H]^{3+}$ was detected, I expect that the signal intensity was much lower than $[M+2H]^{2+}$. Although mass difference between these peaks is 8 Da, since $(633.0342 - 630.3485) \times 3 = 8.0571$, the spectra contain many noise peaks that means low signal intensity. Could you explain why the charge states +3, instead of +2 was used?

Response: We would like to point out that the mass difference in the comment was again miscalculated and the mass shift and peptide shown in the figure are correct. To answer the additional questions regarding charge state: Every peptide will occur in a +2 and +3 charge state and both charge states are commonly observed and detected in a mass spectrometer. Which charge state will be identified in a given experiment is influenced by a multitude of factors such as other ions present in the same precursor scan, ionization and ionization properties, as well as specific settings on the mass spectrometer. In this case, the data shown in the figure represent the best quality spectra for that peptide. We would like to make the reviewer aware that MS1 spectra such as this one in Figure 3B contain of course more than one precursor in any given scan as these are the full survey scans. For easier readability, we have now just included the relevant peptide precursor peaks.

3) It is difficult and impossible to find the raw data files related to Figs.1D and 3B. I suggest to :
1) rename the files, indicate the CFTR constructs (WT, F508del, G551D and N1303K) and the condition used (e.g. drugs etc.), and 2) introduce sub-folders for Fig. 1D and 3B in the data base where the raw data files were placed.

Response: We have now reorganized the raw data files in the repository to clearly label CFTR constructs and experimental conditions (e.g., WT, F508del, drug treatments).

4) The CPP was initiated by adding both formaldehyde and sodium cyanoborohydrate to cell, and the labelling time and temperature was 10 min on ice, respectively. I would like to know the labelling rate at this low temperature to label the fully solvent accessible Lys residues of CFTR in cell. Were the labelling conditions of CFTR optimized?

For this, could you monitor the signals of both labelled and unlabeled peptides by only CH3 with time-course, but no second CD3 labelling? Specifically, I would like to see in which time point the MS signals of unlabeled solvent exposed Lys sites are disappeared.

Response: The CPP labeling conditions for cells were optimized and control experiments were performed such as the one suggested and the results published in Ref.1. We have not observed evidence that the labeling occurs at different rates for different proteins in a cell as the label is membrane permeable, well established and it is known that formaldehyde labeling as a Schiff base reaction is extremely efficient (see Boersema et al., 2013 and references therein).

Considering that most of the detected Lys residues in the WT-CFTR are fully accessible after 10 min labeling, shorter covalent labeling would be more informative for reporting CFTR conformational perturbations. Determining the in vivo labeling kinetics should provide an answer for this concern.

Response. We believe that this comment results from a misunderstanding of the CPP technique. If a lysine is solvent accessible or not is only dependent on its fold and the cellular environment. **The label is merely a tool to measure the solvent accessibility and does not change it.** The epsilon amino group of lysine residues and the N-terminus of peptides and proteins has been extensively studied in bioconjugation because of its nucleophilicity and high surface exposure and most lysine sites are fully solvent accessible in vivo (for review see for example Hermanson. Bioconjugation, 2013). Thus, the results that most detected Lysine residues are fully solvent accessible is fully expected and is consistent with all existing literature. Since the labeling is

complete after > 15 seconds, shorter labeling times will not alter the results, but shorter than 15 s could lead to incomplete labeling and inconsistent data (see also Bamberger et al., 2021).

5) K95 is in the pore of CFTR in both inwardly and outwardly opened configuration according to the available cryo-EM structures (5UAK and 6MSM). In Fig 1B, the solvent accessibility of K95 of WT-CFTR is 96%. I'm not sure if both formaldehyde and sodiumcyanoborohydrate reach K95 and perform the covalent labelling in the pore. Could you verify that these two molecules are able to reach K95?

Response: Only peptides containing either light or heavy labeled 95, but not unlabeled lysine 95 will be identified by ProLuCID, indicating that the probes do reach K95. If the lysine wasn't labeled, it would not be identified in the search. It is also likely that the *in vivo* conformation of CFTR differs from the reported Cryo-EM structures as the Cryo-EM structures contain deletions (such as the RI element) and amino acid substitutions and are of course not in the cellular environment. Furthermore, the conformation of immature CFTR in the ER or Golgi will necessarily differ from the Cryo-EM structures and it is possible that the pore space is different in these conformations as discussed in the manuscript.

7) In p.4, l.136, authors showed that VX-809 changed the solvent accessibility of K1218. This experiment was performed with Δ F508-CFTR. Could you see the same effects on WT-, G551D- and N1303K-CFTR? Could you show if the solvent accessibility change at K1218 by VX-809 is the common effect on other CFTR variants or F508del-CFTR specific effect? If it is the F508del-CFTR specific effect, could you explain why VX-809 change the solvent accessibility of K1218 only for F508del-CFTR, while this corrector has ~not more than 3% rescue efficiency of the mutant processing defects?

Response: Thank you for the comment. Lee et al., 2014 from the Sorscher group first published that CFTR Lys 1218 is critical for CFTR maturation and stability as identified by site directed mutagenesis and that the K1218R mutant increases the amount of total and cell surface wt CFTR. Work by the Lukacs group and others further showed that K1218 is structurally important for CFTR folding. Further work by van Goor et al., 2013 showed that VX-809 decreased the proteolytic sensitivity of NBD2, indicating a more compact conformation of Δ F508 NBD2. Taken together this may explain the effect of VX-809 on Δ F508 CFTR. This is discussed in the manuscript now. As VX-809 does not significantly affect wt CFTR, G551D or N1303K CFTR, the suggested experiment was not performed. However, the observed Orkambi resistance of N1303K may be explained by failure of VX809 to induce a more compact formation of its disordered NBD2 domain.

8) In Table 1, there are 2 Lys sites (K370 and K377) in the peptide DSLGAINKIQDFLQKQEY. In Fig 1B, only the solvent accessibility of K377 is shown in right panel, on the other hand, only that of K370 is shown in the left panel. Likewise, only the surface accessibility of K370 was determined in Fig.2D. Could you explain how did you quantify the solvent accessibility if peptides have multiple Lys sites? If you can separately determine the solvent accessibility of K370 and K377, both Lys accessibilities should be shown in Fig.1B right and left panels?

Response. Corrected. Due to the nature of a chymotrypsin digest, peptides containing either Lys 370 or both Lys 370 and Lys 377 were obtained and both showed high solvent accessibility (>95%). We have now included a separate row for each one in Table 1, whereby the value

presented for the peptide containing both Lys 370 and Lys 377 is the aggregated value for the peptide.

9) Fig. 2C shows the reduced solvent accessibility of K273 of WT-CFTR in the presence of VX-770. Authors claimed the coupling of ICL2 to NBD2 upon activation with VX-770. Can similar results be obtained in G551D-CFTR?

Response. The experiment was done. However, we were unable to identify the respective peptide in the experiments, likely due to lower coverage overall. We therefore cannot conclude if similar results can be obtained in G551D CFTR, but we would speculate that it could be.

Minor concerns:

1) In introduction (p.2, l.62), authors stated that “...several of which are located in the small disordered region, the RI element, is deleted in recombinantly expressed CFTR used for the Cryo-EM studies.”, but many of available cryo-EM structural determinations were carried out on on CFTR channels that contain the RI region, although the disordered region is not visible. This statement could be misleading.

Response. The RI element and the R-region are two different regions of CFTR. Assuming that the comment refers to the R-region, we agree - the R-region is contained in each Cryo-EM structure. However, the RI element is deleted in all of them, as the CFTR structure is not stable with the RI element Accordingly, the sequence for the RI-element is missing in the PBD entries as well.

2) Ref 9 was published in a peer-reviewed journal, on the other hand, ref 13 was published in bioRxiv.

Response. Thank you for pointing this out. We have removed the earlier bioRxiv citation and only included the JPR citation.

3) In p.3, l. 98, “Twenty-six lysine sites in CFTR were quantified in 3 different biological replicates. (Fig. 1B, 1C; Table 1). However, the figure legend indicates: Wt CFTR n=4; Δ F508 CFTR n=4 (biological replicates), p11, l.481. Moreover, in Fig 1B, many data points in each lysine site are shown. Could you explain why there are many data points although biological replicates were 3 or 4? Please explain what the error bars are.

Response. Thank you for pointing this out. Four different biological replicates were acquired for both wt and Δ F508 CFTR. We have corrected this in the main text. Because a peptide can be measured more than one time per replicate, more than 4 measurements for each site may be present. These measurements may represent partially different peptides containing the same site and different chromatography steps per replicate, increasing the confidence for the site quantification. The error bars reflect standard error of measurement. This is now indicated in the figure legend.

4) In Fig 1B, 2B and 2D, F508 should be F508del. Response. The F508 deletion is indicated as Δ F508 in each figure. We have saved the file now in a newer Acrobat format, so this is visible.

5) In p.4, l.151, PDB 5UAR is the cryo-EM structure of zebrafish CFTR with dephosphorylated, ATP-free states. Please use the right PDB file to determine the solvent accessibility of K273 in human CFTR. Also, ref 22 is the literature of zebrafish CFTR dimerized forms. Please cite the literature of human CFTR dimerized forms.

Response. Thank you for catching this typo. We compared the solvent accessibility of both inactive and activated human CFTR (PDB 5UAK, and 6MSM). It is now corrected in the text.

6) l.318 Structural differences not only in the NBD2 but all structured domains of the F508del-CFTR have been documented in relation to that of the WT-CFTR by several groups, please include appropriate references.

Response: We are well aware that many groups have performed limited trypsin digestion to assess structural differences between wt and Δ F508 CFTR in various different settings, primarily for NBD1. However, we were just emphasizing that others had observed differences in the conformation of NBD2 between wt and Δ F508 CFTR, and we cited the first paper that identified differences in NBD2 between wt and Δ F508. We have now added more references.

7) Could you please clarify the meaning(s) of following sentence. l. 327: Point mutation of K1218 to K1218R has been shown to increase the amount of CFTR, stabilize it at the plasma membrane and reduce lysosomal degradation of CFTR, while its deletion improves folding efficiency of Δ F508 CFTR.

Response: Lee et al., 2014 showed that the K1218R point mutation stabilizes CFTR at the plasma membrane and reduced the lysosomal degradation of CFTR. Du and Lukacs showed that deletion of K1218 improves folding efficiency. We now discuss this in more detail.

Reviewer #2 (Remarks to the Author):

Summary:

Pankow and colleagues present an application of their Covalent Protein Painting (CCP) method (Bamberger et al., 2021 J. Proteome Research), to the challenge of understanding the conformational landscape of Cystic Fibrosis Transmembrane Regulator (CFTR). As a protein of therapeutic interest, CFTR has a range of well-characterised conformational variants. The authors exploit the abundance of structural data, model cell systems, and chemical/physical modifiers available for CFTR to contextualise their CCP findings and reveal novel conformations of CFTR abundant in the cell. While this is a nice showcase of the methodological capabilities, the importance of these findings for a broad readership is unclear. Unfortunately, critical weaknesses in, and the description provided for, several data visualisations presented throughout the manuscript prevent these results from being adequately examined and interpreted. The authors also do not provide robust experimental controls for the tagged CFTR variant work and essential methodological information is missing which would preclude replication of the study as a whole. Together, I believe these issues render the work as presented unsuitable for publication in Nature Communications.

Response. We appreciate the comments of the reviewer and have addressed each concern to improve the manuscript. We have now included more detailed descriptions of the data and have indicated statistical significance in graphs where appropriate. A summary data table (Supplemental table 2) lists all lysine accessibilities by residue and experimental condition. We hope that these efforts will make the results easier to read and interpret, as individual data points required in the plots indeed makes readability more difficult. We have now also added a sentence in the main text and in Figure 1 that explains that each point in the graphs for the solvent accessibility represents a measurement for an individual peptide containing the specified lysine site. As more than one peptide may contain a specific lysine site due to the chymotryptic digest and peptides may occur in more than one charge state, more than one measurement per biological replicate may be obtained. Regarding the tagged CFTR work, we have included empty vector controls as direct controls and have compared all data to the previously published interactome data set of CFTR, which was acquired in the same way. Our responses are detailed below.

Major concerns:

1. Data presented in Figures 1B, 2B, 2D, 3A, S1A, S1B: Each of these panels is a variation of scatterplot visualizing the accessibility of specific residues under a given condition. While the authors' effort to include individual data points is acknowledged, the plots are extremely challenging to interpret. The data summary elements (mean, SD/SEM?) often cannot be distinguished from the replicate data points, and the results referred to in the text as having significant differences are not annotated. The overlap in different treatments (especially Figure 2D) makes differences between the treatments exceptionally difficult to evaluate. In addition, the link between the number of replicates specified in text, in the figure legend and individual data points shown on the graph is unclear. For example, in Figure 1B, $n=3$ in the main text, $n=4$ in the figure legend, and the actual plot shows some lysines with upwards of 10 individual markers in a single experimental condition. One assumes these might come from quantification of different peptides containing the lysine of interest however this is not described in the legend or methods. In addition, the summary elements are not described in the legend making it unclear if the error bars represent SEM/SD or something else, and therefore it is not possible to rationalise why some error bars are unidirectional while others are present both above and below.

Response. Thank you for this important feedback. As suggested, the number of data points can reflect multiple peptides per lysine site due to different charge states and peptide fragments generated by the chymotryptic digest. We have now clarified this in both the main text and the figure legends. We checked if any error bars were unintentionally set to unidirectional, but this was not the case and we believe that some error bars were plotted erroneously as unidirectional by the software due to plotting space limitations. We have now used a newer version of this software. Additionally, we have enhanced the figures by clearly annotating statistical significance and error bars (mean \pm SEM). We have kept the individual data points to comply with the journal recommendations but agree that this makes the data challenging to read. A summary data table (Supplemental Table 2) where mean and SD values for each peptide are given side by side for each peptide and each condition is included, so that values can be more easily compared. We hope these improvements help make the results more accessible.

2. Experimental details concerning the cell lines, and the rationale behind the use of each of the different cell lines throughout the course of the study, are lacking. How are the CFTR variants expressed in the CFBE/HBE cells? Why is the K273 mutation then profiled via transient

overexpression in a non-specialist cell type (HEK293)? Different cell models likely impact the conformation and available interactome, do the authors expect their results to be generalisable to the specialist cell types?

Response. We appreciate the insights regarding cell line choice. HEK293 cells, while not airway-specific, have been widely used to study CFTR function due to their epithelial origin and preserved CFTR function. Several landmark experiments that established CFTR function and turnover were carried out in HEK293 cells (e.g., Ward and Kopito, 1994; Domingue et al., 2014). We followed this precedent in profiling the K273 mutation. The CFBE410- and the HBE410- are isogenic cell lines derived from patients that express either Δ F508 CFTR or wt CFTR, re-inserted into the original locus by SFHR-targeting (Bruscia et al., 2002 Isolation of Isolation of CF cell lines corrected at DeltaF508-CFTR locus by SFHR-mediated targeting). These cells however cannot be efficiently transfected and it is common practice in the field to use other cell lines for other mutations or overexpress them transiently in HEK293 cells as we did here. Similarly, FRT cells expressing different CFTR variants are commonly used in the field. While minor differences may exist, function and turnover of CFTR and misfolded variants appears to be very similar between the cell lines as evidenced by a large body of publications. A brief overview of the usability of cell lines for developing CF therapeutics is provided by Clancy et al., 2019. We have now added a Supplemental Figure 4, which shows that solvent accessibilities for wt CFTR are very similar in HEK and HBE410- cells. Furthermore, the core interactome data collected for both wt CFTR and Δ F508 CFTR in HBE410-and CFBE410- cells does not seem to differ significantly from the core interactome for K273A CFTR in HEK cells, supporting the generalizability of our results across cell types. Finally, supplemental table 2 also shows that in general solvent accessibility of residues is similar between different cell types and conditions for the majority of residues.

3. Figure 1B: There appear to be substantially fewer lysines quantified in the label swap experiment - why is this? Does this point toward an inherent bias in the method whereby one version is favoured for best coverage? Similarly, some lysines appear to be quantified only in one of the cell lines, were these peptides not identified in any of the replicates for that cell line? How does this affect the conclusions made about these residues with respect to solvent accessibility?

Response: The label swap experiment primarily ensures labeling consistency, which was confirmed in these experiments. Although there are fewer lysines identified, this does not influence the quantification and is due to fewer replicates in the label swap experiments (n=2), which is now indicated in the figure legend. The quantification in CPP is not affected because each ratio is measured in the exact same sample simultaneously for heavy and light isotopes. Once a peptide has been identified, the quality of the ratio measurement is independent of the peptide identification. There are separate parameters and quality requirements for quantification of the MS1 peaks that will generate the ratio and we set very strict requirements regarding peak shape, intensity of the precursor ions and other technical parameters as mentioned in the methods section to make sure that ratio measurements are accurate.

4. Figure 2: The legend for panel A only discusses one drug but the heatmap appears to present several combinations. There also appears to be the cell lines alone (WT, d508) but it is not clear in the legend or figure whether the treatments are being conducted on the WT or mutant background. In addition, the colorbar labels are not adequately sized or specific (what 'value' is shown?). Similarly, panel D includes the following legend descriptors: wt, wt_VX770, F508, Trikafta, VX445-VX661, F508_28 celsius. This does not make sufficiently clear which conditions

are applied to which cell lines. It is not clear to which comparisons the annotated p-values refer to, nor the other annotated summary measures.

Response: We sincerely apologize for the oversight. We accidentally included a figure legend for an older version of Figure 2 and have corrected the issue now.

5. The immunoprecipitation experiment (Fig. 4C, D) uses GFP as the control protein against which the interactome of the K273 is determined, however, this does not discriminate native interactions maintained by the variant from those specific to the conformation modelled by the K273 mutation. A WT comparison is required to understand the specific interactions associated with the novel conformation explored in this study.

Response: We agree with the reviewer and have compared the interactome of the K273 mutant to the interactomes of wt CFTR, Δ F508 CFTR and CFTR null cells as well the GFP control. This is now mentioned and made clear in the main text and the figure legend.

6. Figure 4: The network interaction map provided in Figure 4C is illegible at standard zoom and not adequately described in the legend (e.g. what do the directional arrows represent?). Similarly, how were the proteins presented in Figure 4D selected from the list of interactors – what does ‘specifically enriched’ mean? Were they the most significantly enriched? And were all the proteins presented identified in the interactome? This appears to be a protein-protein interaction map, what database was used? Was any filtering for confidence in those interactions completed? These details are missing from the methods. The legend for this figure is similarly lacking in important details, for example how the protein nodes are colored and sized, and by what mechanism the proteins presented were arranged/clustered. Together these issues prevent accurate interpretation of the data.

Response. We have now included a more detailed description of the network interaction map and related methodology. Briefly, the network map is a map of statistically overrepresented GO terms in the K273A mutant as compared to wt CFTR and controls. GO Analysis was performed as described for GO Miner and is based only on the identified proteins in the interactome. Interactors were filtered for confidence with a P-value of greater than >0.9 in CoPIT generator^{13,33}. Statistical overrepresentation of GO terms in the identified interactome was assessed with BiNGO in Cytoscape 3.8.2 using corrected p-values as described in Maere *et al.*, 2005. BiNGO also generates the network of overrepresented terms in Cytoscape, whereby the area of a node is proportional to the number of genes in the experimental set annotated to the corresponding GO category. Corrected P-values for statistical overrepresentation of a GO category in the dataset are indicated by color according to the color scalebar. The arrows represent directionality between categories as far as we are aware. Interactions between identified interactors in the ESCRT complexes and the retromer complex were identified and visualized with the GeneMANIA 2.2 Plugin in Cytoscape 3.2.8 using physical interactions reported in BioGRID-small scale studies, BIOGRID and BIND as well as Pathway information in Pathway Commons. We have now included a description in the methods sections and the figure legends.

Minor comments:

1. Data tables as provided in the supplementary material are in a pdf-embedded format. Please provide a format more suited to tabular data e.g. csv to improve the openness and reusability of the dataset.

Response. Datasets are now included as csv files.

2. Discussion of the importance or relevance of these findings to the broader audience, or even to the CF field where one imagines the novel conformation is of most interest, is limited.

Response. Thank you for the suggestion to contextualize our findings more. We have expanded the discussion to emphasize the broader relevance, especially in the CF field, and to underscore the potential therapeutic implications of these novel CFTR conformations as possible given the space limitation.

3. Figure S2 contains no evidence of magenta staining at all. Is general diffuse staining expected (e.g. for LC3, as observed in HeLa)? And if not, a positive control is required to demonstrate successful staining and prove the absence of reactivity in the cell lines of interest. In addition, the legend indicates a comparison with WT cells but the panels presented are both labelled K273.

Response. The negative result is expected and consistent with current literature. While autophagy markers LC3 and SQSTM are generally not expected to be present in normal cells (see also product datasheet for the antibody), cells expressing $\Delta F508$ CFTR and other misfolded CFTR variants have particularly impaired autophagy (see Luciani et al, 2010 for example).

4. Line 212: “Appears to be Golgi-related structures”. This statement is not supported by immunocytochemistry, only the absence of autophagy markers. Excellent markers exist for the Golgi compartments.

Response: This statement is based mainly on the interactome data. We have now incorporated Golgi-specific markers such as Golgi red and have updated the manuscript accordingly.

5. Method descriptions –brief method outlines should be preferred over references to aid reproduction of the method as applied in this specific study (line 362, Line 367, Line 375, line 380, line 427).

Response: We agree with the reviewer and have kept references to a minimum, but due to the complexity of the methods and the fact that a whole Nature Protocols paper exists describing every step involved in the CFTR immunoprecipitations in detail as well as the data analysis methods, we felt that the mentioned references were appropriate here, especially as it is important to follow all the steps in the protocol to reproduce results.

6. Readability – extended sentences and grammatical structure should be revised, for example, lines 40-44, lines 128-136.

Response. Thank you for noting specific sections in need of revision. We have simplified extended sentences throughout, particularly in lines 40-44, 128-136, and other sections as recommended.

7. Line 120 – 121: Comparing solvent accessibility with solvent inaccessibility makes interpreting this result more challenging than necessary. Please compare apples with apples, e.g. “Solvent-accessible in only 60% of wt CFTR molecules, but to greater than 95% in $\Delta F508$ CFTR”.

Response. We have revised this comparison for clarity, now stating, “Solvent-accessible in only 60% of wt CFTR molecules, but in greater than 95% of $\Delta F508$ CFTR”.

8. Descriptions for each of the cell lines used should be provided in text at first mention, followed by the abbreviation to be used throughout the remainder of the manuscript. Line 89 human bronchial epithelial cells expressing wt CFTR (HBE16o-) abbreviated later to HBE, Line 107 – isogenic CFBE41o- cells expressing Δ F508 CFTR abbreviated in methods to CFBE. Similarly for other abbreviations used throughout the manuscript (e.g. GO is not defined).

Response. We have ensured that each cell line and abbreviation is now clearly defined upon first mention. Additional abbreviations, such as GO, are now also introduced.

Reviewer #3 (Remarks to the Author):

Thank you for providing your input. We hope you had an enjoyable learning experience.

Reviewer #4 (Remarks to the Author):

This manuscript describes the application of a protein foot-printing approach using lysine labelling to characterize CFTR conformational changes associated with various Cystic Fibrosis disease causing mutations in CFTR as well as with various drug treatments. The so-called covalent protein painting (or CPP) approach used in this work has been previously described and used in other applications by the authors. One attractive and unique feature of CPP is that it can be used to study proteins in vivo, which is done in the current manuscript. Another important feature of covalent labelling strategies like CPP is that they can capture “dynamic” information about protein structure that is missed using other structural biology techniques like x-ray crystallography and Cryo-EM. In fact, this is demonstrated/highlighted in the current work where a unique CFTR conformation involving the ICL2 and NBD2 domains of CFTR is detected. As part of the work described in the current manuscript the authors investigate the disease relevance of this conformation, which can be specifically probed by measuring K273 protection in CFTR. Their studies on various disease mutants and drug treatments reveal that the newly discovered solvent-exposed K273 conformation of CFTR is an inactive conformation that is also observed in deltaF508 and N1303K CFTR mutants. The authors’ CPP data also provided for a better understanding of VX-809 drug mode-of-action.

Overall, the work is well-down and well-presented, with a few exceptions noted below. The findings are not only significant from a technique development perspective (e.g., the demonstration of CPP unique ability to detect CFTR conformations not previously captured using other more conventional structural biology methods), but it is also significant from a CFTR disease biology perspective (e.g., the new details that are learned about VX-809’s mode of actions, and the newly discovered presence of a biologically relevant protein conformation in CFTR). Thus the work stands to be of great interest to the general readership Nature Communications, where it would be appropriately published (after minor revision-see below).

Other more specific comments that need to be addressed before publication:

Line 31: should be “identifies”

Line 90: not clear why “(CPP)” is placed at the end of the sentence? Maybe it should read, “...according to the CPP workflow in Fig 1A.”

Line 124: Should include references to the Cryo-EM structures.

Line 200: delete “well as”

Lines 310 and 320: The last 2 paragraphs both begin with “Finally”. Probably just the last paragraph should begin with “finally.”

Refs 9 and 11 seem to be to the same paper. Maybe just used the peer-review publication in JPR (ie., ref 9).

Response. We would like to thank this reviewer for the constructive and very helpful comments. We have corrected all specific concerns above as requested and addressed the concerns below.

In the Figure panels showing the CPP solvent accessibility data, it is not clear what the actual points represent. Seems like these should be the data from the biological replicates? But this does not seem to be the case because the numbers don't match up. This needs to be better explained in the legends.

Response. We agree with the reviewer that this can be confusing. We now explain in the main text and in the figure legend that each point in the graphs represents a measurement for an individual peptide containing the specified lysine site. Since multiple peptides may contain the same lysine due to the chymotryptic digest and can appear in different charge states, multiple measurements may be obtained for a single biological replicate and thus there may be more points than number of replicates.

Figure 2 is confusing especially since it contains 4 panels but the legend only describes 3. And the existing description of the first three panels is confusing (almost to the point that it doesn't seem to be describing the existing panels). This needs to be fixed/clarified.

Response. We would like to thank the reviewer for catching this error. The previous figure legend represented an older version of Figure 2 that had been replaced. We have now updated the figure legend to reflect the updated figure panel and sincerely apologize for the confusion.

Reviewer #2 (Remarks to the Author):

Pankow and colleagues present an application of their Covalent Protein Painting (CCP) method (Bamberger et al., 2021 J. Proteome Research), to the challenge of understanding the conformational landscape of Cystic Fibrosis Transmembrane Regulator (CFTR). As a protein of therapeutic interest, CFTR has a range of well-characterised conformational variants. The authors

exploit the abundance of structural data, model cell systems, and chemical/physical modifiers available for CFTR to contextualise their CCP findings and reveal novel conformations of CFTR abundant in the cell. While this is a nice showcase of the methodological capabilities, the importance of these findings for a broad readership is unclear. Efforts to add detail to the methods and figure legends have improved the manuscript substantially. However, several aspects of the previous review are yet to be addressed and clarification is required to confirm the data supports the manuscripts conclusions.

1. Major Concern #3 remains unresolved. The authors state “The label swap experiment primarily ensures labelling consistency.” (rebuttal), and “A label swap experiment confirmed the detected differences between wt and Δ F508 CFTR” (Line 116). However, some residues disagree in solvent accessibility by up to 10% between the two orientations, and there are differences in whether the residue was quantified in only one or both of the cell lines. Importantly, this difference appears in one of the residues highlighted by the authors (K95), where the Light-heavy F508 is ~76% (compared to WT 97%) and the Heavy-light F508 is ~86%, while the Heavy-light WT is not quantified. This example highlights that not all peptides are consistent, and reinforces the importance of two questions not addressed by the authors in this revision: *Some lysines appear to be quantified only in one of the cell lines, were these peptides not identified?* and *How does this affect the conclusions made about these residues with respect to solvent accessibility?* To elaborate, the authors state that “Both sites were less accessible in F508 CFTR” (line 123), however, it is unclear how the effect of the F508 mutation can be described in this manner without the WT measurement to compare to.

Response: The intent of this label swap experiment is only to show that the trend and differences are similar in general when labels are switched. We do not expect the values to be exactly the same for the label swap experiment, as it is known that deuterium labeling can slightly shift chromatographic retention times, and thus influence quantification. Small differences such as the 10 % difference mentioned could be expected, and are often seen in other label swap experiments from other groups, even though we agree that it would be nice to have the measurement for this particular wt peptide. In this particular case though, the peptide was not identified for the heavy -light labeling in wt, only in F508. This is a very common occurrence in discovery and data dependent proteomics approaches, and it is extremely unlikely and uncommon that all peptides will be identified in all experiments.

The reliability of the measurements for this peptide may be gleaned from other data points collected for this peptide as well, as the measurements between different conditions such as F508 treated with DMSO or VX809 or at 28°C were very similar to the 77% measured for F508 and within the standard error of mean, see table below

Lys	condition	Percentage solvent accessibility
K95	F508	77.03%
K95	F508 (0.01% DMSO)	80.65%

K95	F508 (10 millimolar VX809)	79.97% +/-3.87, n=10
K95	F508 (28 °C)	83.78 % +/- 4.17, n=3

2. “We checked if any error bars were unintentionally set to unidirectional, but this was not the case, and we believe that some error bars were plotted erroneously as unidirectional by the software due to plotting space limitations. We have now used a newer version of this software.” (rebuttal). However, Figure 1B still contains unidirectional error bars (Light-heavy) and lacks error bars entirely (Heavy-light), and Supplemental Figure 1 contains unidirectional error bars.

Response: We apologize for the oversight. All of these three figures now have error bars in both directions.

3. “We have now clarified this in both the main text and the figure legends... Additionally, we have enhanced the figures by clearly annotating statistical significance and error bars (mean \pm SEM).” (rebuttal). Figure 3A contains error bars that are not described in the legend.

Response: Thank you for pointing this out. We have added the description.

4. The decision to use different cell lines is now well justified (provided in response to Major Concern #2). Please include this information in the manuscript as it provides important context required to interpret the presented results for readers from a general audience.

Response. Thank you. The justification is now incorporated into the discussion section.

5. The labelling for Figure 2A is still unclear. The legend states “Heatmap displaying solvent accessibility changes of Δ F508 CFTR upon treatment of CFBE41o- cells with Trikafta (24 h), the two corrector compounds VX-661(5 μ M, 24h) and VX-445 (3 μ M, 24h), corrector VX-809 (5 μ M, 36h), at permissive temperature of 28 °C (24h), or in DMSO solvent controls and of wt CFTR.” (line 513). However, the labels on the heatmap and scatterplot contain “F508”. How is this different to the DMSO solvent control (labelled DMSO)?

Response: Δ F508 in Figure 2A is an additional control not treated with any compound or solvent to be able to distinguish possible effects of the DMSO solvent on Δ F508 CFTR conformation. This has been added to the figure legend, also for the scatter plot in Figure 2B.

6. Figure 2D remains uninterpretable owing to overlap in different treatments. While the means are now provided, the authors should consider alternative visualisation options e.g. individual scatterplots for each lysine residue where the different treatments can then be visualised as separate columns.

Response. The point of figure 2D is that most treatments do not lead to the same conformational changes of CFTR seen when wt CFTR is activated with VX-770 and the solvent accessibility stays the same as in the untreated condition. Hence, the extensive overlap between treatments: “However, the coupling of the ICLs with the TMDs and R-region changes were not accompanied by the same changes to residues K464 or K370 involved in channel gating and the ion permeation pathway as in wt CFTR. Instead, solvent accessibility remained similar to that of untreated or solvent control treated Δ F508 CFTR (Figure 2D).” We think that the figure illustrates this and how only VX-770 leads to substantial changes well. We have scaled the y-axis now differently and have indicated the means in the color corresponding to each treatment.

7. Figure 4C: Text size for the node labels remain unreadable. More importantly, components of the visualisation are still not adequately described. The authors state “The arrows represent directionality between categories as far as we are aware.” (rebuttal). What is directionality i.e. what does the direction of the arrows represent when linking two nodes? They appear to link hierarchical elements of the ontology, but this is not described. This needs to be clarified.

Response: Yes, arrows indicate hierarchy of the ontology terms according to the GO database. This has been included in the figure legend. We have now also included the Cytoscape file of the network, so it can be viewed and labels read conveniently.

Additional concerns:

8. The inclusion of biological information is commendable. However, it raises a question regarding the validity of statistical comparisons. The actual n for each residue is a combination of biological replicates and the number of distinct peptides containing that residue, which can vary significantly (some have 1 data point, some > 10). Since increased n can correlate with a higher likelihood of detecting significant results (Shreffler & Huecker. “Type I and Type II Errors and Statistical Power.” StatPearls, 2024), could the authors clarify how they address the potential confounding effects of combining replicate and peptides to artificially increase n? This method also obscures variability between biological replicates/conditions, potentially affecting the interpretation of visualisations.

Response: This is a common concern in proteomics experiments and we have addressed this using PCQ and SoPAX, which were developed for such situations and use several different statistical approaches such as Benjamini Krieger Yekutieli correction for example (Bamberger et al., 2018, Bamberger et al., 2021).

9. Several observed differences (e.g. Figure 1B) appear to be based on data from a single peptide within a single replicate (one data point). Please elaborate on the reliability of these differences, particularly given they are detected in only 1 out of 4 biological replicates - how does this impact confidence in the findings for these data?

Response: We went above and beyond to confirm the differences for these peptides and manually validated them for intensity, peak shape and ID- which is still the gold standard in the field. We set very strict quality filters for identification of peptides and for quantification in Census to filter out any peaks with questionable ID, intensity and peak shape in general.

10. Supplemental Figure 4 is a table (not a figure)

Response: Corrected. It is now Supplemental Table 3.

Reviewer 1

1. It is not clear which CFTR variant “conformational changes” were monitored “during its biogenesis”. I could not find any methodology in this manuscript that can monitor either the cotranslational or the posttranslational conformational biogenesis of any of the CFTR variants.”

Response: The information that the conformational changes we observed occur during biogenesis are supported by several facts and results in the manuscript and are discussed in the discussion section. It is well established that > 90 % of Δ F508 CFTR in CFBE 410- cells does not reach the plasma membrane as evidenced by its absence of function and the lack of a fully mature “band C” protein. The Δ F508 CFTR protein that is nevertheless expressed and translated is mostly located in the ER (and contiguous compartments) – being folded, refolded, transported, maturing, retrograde transported and degraded – in short undergoing biogenesis (reviewed and defined for example in Hegde et al., 1999, Cell as: “The biogenesis of most secretory and membrane proteins involves targeting the nascent protein to the endoplasmic reticulum (ER), translocation across or integration into the ER membrane and maturation into a functional product”.) Thus, the conformation seen in untreated Δ F508 CFTR cells reflects mostly the conformation of Δ F508 CFTR in the ER undergoing biogenesis. The same is true for N1303K CFTR. That the K273 “conformational defect” is seen in both mutants to the same extent (although more N1303K CFTR is transported to the plasma membrane) and it is also seen to a lesser extent in wt CFTR and G551D CFTR argues that this a general conformation all of these CFTR variants undergo- despite their different functionalities and different incorporation into the plasma membrane.

2. “Then wesuggest possibilities to stabilize misfolded CFTR variants not responsive to these drugs such as N1303K CFTR.” This is incorrect, as published preclinical and clinical studies demonstrated the responsiveness of the N1303K CFTR to available pharmacophores (e.g.PMID: 34379998 38226069, 38463894.)

Response: It is generally accepted that N1303K CFTR is minimally responsive to Trikafta with low absolute responses (Veit et al., 2020). We have altered our abstract to say “minimally responsive”. We would like to note that it is a general consensus in the field that better drugs still need to be developed and there are active programs to do so (see CF foundations website for example).

3. No functional evidence is provided to support a novel opening mechanism in the presence of VX-770 without activation of the PKA (e.g. by forskolin). The assumption that the channel is sufficiently phosphorylated constitutively should be demonstrated as its extent cell-type and condition dependent. Experimental evidence suggests the contrary as detailed in point 5 (here point 8).

Response: See point 8 for detailed response.

4. “While the increased accessibility of the K273 of the F508del-CFTR is undisputable, I have concerns regarding the significance of this observation in the light of extensively documented conformational defects of the TMD1, TMD2, and NBDs of the F508del-CFTR mutant in previous publications. These studies measured the protease susceptibility of proteolytically generated domains (e.g. in the Domain Interdependence in the Biosynthetic Assembly of CFTR, J. Mol. Biol. (2007) 365, 981–994, Fig.7). Thus, recognizing a single residue increased labelling efficiency in the TMD1 does not add substantially novel information to our understanding of the F508del-CFTR conformational defect.

Response: The results in the cited manuscripts have all been performed with recombinant CFTR or in vitro. This is the first in vivo study that can point out structural differences on a residue level. That the ICL2 is completely solvent accessible in Δ F508 CFTR, N1303K CFTR and non-activated wt CFTR, but binds to its pocket in NBD2 upon VX-770 (or Trikafta treatment,

but not VX-445 or VX661 treatment) has not been captured in any Cryo-EM structures, yet appears to be a general mechanism required for activation.

5. It is not clear what measures were taken to ensure the structural integrity of CFTR during the extensive dimethyl labeling. Covalently attaching 52 methyl groups or may be more to CFTR changes the side chains hydrophobicity and most likely perturb the protein dynamics and tertiary structure so that any subsequent modification may no longer reflects the protein's initial structure. Providing appropriate control experiments to ensure that the structural probe (dimethyl modifications of Lys residues) does not imposes structural perturbations is even more critical for CFTR mutant variants that are conformationally unstable and, therefore, more susceptible for structural perturbations during labelling. Demonstrating the structural integrity of labelled CFTR variants would be essential for validating the methodology. As the authors did not provide any evidence that the structural (see also Mendoza VL, Vachet RW. 2008. Protein surface mapping using diethylpyrocarbonate with mass spectrometric detection. *Anal Chem* 80:2895-2904).

Response: Based on the reviewers sentence "that any subsequent modification may no longer reflects the protein's initial structure", we have the impression that the technique is still not clear and would like to clarify that no such modifications are carried out after the first labeling step, which takes about 15 seconds or less, nor are protein dynamics captured. We have extensively demonstrated the validity of the method in previous papers with data on >10,000 proteins that agree with PDB data on solvent accessibility. If the method would randomly disturb the structure of CFTR or any other protein, consistent results would not be expected nor consistent agreement with crystal and Cryo-EM structures, yet both is the case. Regarding the DEPC labeling, Dr Vachet tests for structural perturbation because diethylpyrocarbonate is a much larger modification, modifies the protein more extensively (up to 30% of the protein), has different properties and -unlike CPP- labels proteins in a local environment dependent manner. It should be noted that even for DEPC labeling "detectability at higher extents of labeling and MD simulations suggest minimal structural perturbations" (Kirsch et al., 2024).

6. For the biochemical and functional detection of rescued CFTRs, cells are exposed for 24-48 hours to low temperature. This condition ensures sufficient time to accumulate detectable amountof mutants in post-ER compartments because the conformational rescue is inefficient and partial, as well as membrane trafficking is slower at reduced T. Thus, the long incubation time of cells is irrelevant for the dynamical and conformational responsiveness of CFTR during the labeling. Importantly, cooling down the cells not only instantaneously rigidifies the phospholipid membrane (by shifting the temperature below the phospholipid melting temperature, but also rearrange CFTR side chains' atomic motion dynamics and backbone dynamics, thus the interdomain coupling energetics, i.e. the entire conformational space of CFTR. Therefore, the covalent labelling obtained results at 4C do not reflect the "native state in vivo" as claimed by the authors. To support the authors' inference, the CPP studies should be performed at or close to the physiological temperature in both corrector rescued and non-rescued mutants as we proposed previously.

Response: The cells are placed in dish on ice at room temperature in around 20 ml of PBS or media. Cooling cells down to 4 °C or even 0 °C will take at the minimum 2-7 minutes according to Newton's law of cooling, even if assuming efficient heat transfer (which the polypropylene dish does not). Stiffening or phase transitions of the phospholipid layer (in isolated lipid particles nonetheless) is also measured in minutes, see for example Chen et al, 2018, *Scientific Reports*. The maximum 15 second labelling time seems insignificant in this regard.

7. Particularly relevant findings are the Cys-crosslinking of the Cys276 (CL2) with Q1280C and K1284C substitutions in the NBD2 of CFTR in intact cells. Disulfide bond formation between cysteines at positions 276 and 1280 also occurred spontaneously, demonstrating the close contact of the NBD2 and CL2 interfaces in vivo by the Riordan lab. PMID: 18305154

Response: We have now included this reference.

8. This argument is incorrect. Electrophysiological studies in the quoted PNAS paper show that the VX-770 potentiator was unable to activate the G551D-CFTR in the absence of PKA-stimulated phosphorylation (see Fig.1 panel E). Robust current was only provoked after forskolin (Fsk) stimulation of the PKA-dependent phosphorylation. Thus, the VX-770 triggered channel activation is suboptimal, and does not reflect the conformational changes associated with CFTR maximal activation as also stated in the abstract. Similar results were obtained later, by the Bear lab: The CFTR potentiator VX-770 (ivacaftor) opens the defective channel gate of mutant CFTR in a phosphorylation-dependent but ATP-independent manner. J Biol Chem 287: 36639–36649, 2012. Likewise, the WT-CFTR very limited activation by VX-770 was demonstrated in the absence of phosphorylation (e.g. PMID: 39373784).

Response: We believe it is important to distinguish between stimulated CFTR channel **activity** (as measured in an Ussing chamber experiment for example), and structural re-arrangements required for **activation** that take place upon treatment with VX-770. In this regard it is interesting that VX-770 does activate CFTR independent of cAMP as suggested in the reference cited by the reviewer (Castanier et al., 2024), even if the channel activity may be less than with cAMP, but it is present. So, even if the triggered CFTR channel activity by VX-770 is sub-optimal, that does not say anything about general requirements for activation. As discussed in the manuscript, we show very clearly that the inner part of the ICL2 loop where K273 is located does not engage with its NBD2 pocket when the CFTR protein is not activated, as is the case with $\Delta F508$ CFTR, $\Delta F508$ CFTR treated with just correctors or N1303K CFTR or around 60% of wt CFTR protein in unstimulated cells (consistent with the estimated proportion of wt CFTR being active). However, ICL2 very clearly engages with NBD2 in both wt and G551D CFTR variants responsive to VX-770 and upon Trikafta treatment of $\Delta F508$ CFTR, (but not upon VX-661 and VX-445 treatment) and appears to be a requirement for activation. This is also discussed in detail in the discussion section of the manuscript and we do not make claims about maximal activity.

9. The resolution of Fig. S2e images is rather poor, particularly in panel E. The WT CFTR should be included as control. The WT-CFTR cellular distribution pattern is concerning (Fig.4). The WT colocalization with the plasma membrane marker (WGA) is not recognizable. Likewise, the ER colocalization of the K273A mutant is barely detectable. The quality of the images should be improved and scalebars should be included.

Response: We have now included scalebars for the images in Figure 4, which were accidentally hidden behind the images. The point of these figures is to show the clearly different cellular distribution between the wt and the K273A point mutant, which is supported by the interactome results. We have now also included inserts in the pictures that make it easier to see the localization of the GFP tagged wt CFTR at the plasma membrane and the failure of K273A CFTR mutant to localize to the plasma membrane, which instead is localized to the ER/Golgi.

The ER co-localization of K273A CFTR is clearly shown in Figures S2 A-D. We can move these figures to the main figures. Supplemental Figure 2E is from a live imaging experiment with a live Golgi/ER stain and the zoomed in image is therefore fuzzy. We do believe that it is possible to see a partial Golgi colocalization though and we have pointed it out with arrows. The distribution of the GFP signal from wt CFTR in the transfected HEK cells is consistent with results from other groups (see for example Lee et al., 1999 JBC, jbc.274.6.3414).

9. Additional concerns:

Some of the comparative analyses of covalent labelling lack statistical significance analysis (e.g. Fig.1B, Fig.3A). Please include.

Response: We apologize for the oversight. We have corrected this issue for Figure 1B. For Figure 3A, statistical analysis is only appropriate for the comparison between N1303K and Δ F508 CFTR and there is no statistical significant difference between them.

10. My concerns regarding the CPP methodology have not been properly addressed. According to the Method section and Figure 1A, the mass difference between light and heavy labeled peptides should be 8.04 Da. However, the mass difference in Figure 1D is ~4 Da ($\Delta m/z = 943.016 - 941.0068 = 2.0092$ and $z = 2$, and then $\Delta m = 2.0092 \times 2 = 4.02$ in WT.). The authors' response indicates that the "mass difference is 4.02 Da as correctly displayed", so this is not miscalculating. My question was "why the mass difference (Δm) is not 8 Da (as indicated in the Methods), but 4 Da?", and this was not addressed. Is this because of incomplete labeling? (e.g. mass difference of single methylation for both light and heavy would be 4 Da). How about other peptides if their mass difference is not 8 Da, but 4 Da? Could you please, explain why the mass difference was 4 Da in Figure 1D?

Response: The methods section describes the theoretical mass difference between the peptides as 8.04 Da per convention for a charge 1 peptide. The displayed peptide has a charge 2 and thus the mass difference is 4.02 Da.

11. The deletion of noise peaks from mass spectra is not appropriate without explicitly stating this modification. This should be indicated in the Figure 3B legend and shown in the raw mass spectrum in the Supplementary Information.

Response: Noise peaks are not removed from the mass spectrum in Figure 3b, but the full mass range is reduced to a 4 amu wide window encompassing the m/z values used in the quantitation and thus it appears there is no noise. For clarity and to avoid the confusion, we included only this extracted MS 1 spectrum for the identified precursor ions, displaying their relative quantities. This is an accepted standard practice in the MS field and there is nothing inappropriate about this. We are happy to re-include the original figure showing the entire MS1 spectrum for the selected mass range in the supplement (see Supplementary Figure 4).

REVIEWER COMMENTS

Reviewer #2 (Remarks to the Author):

The authors' efforts to incorporate the feedback from the previous rounds of review are acknowledged and appreciated. Please find below some final comments relating to the presentation of the data in the updated manuscript:

1. Figure 2D: The y-axis label is overlapping and needs to be adjusted.

Response: Thank you for pointing this out. Done.

2. Supplemental Figure 3: Scale bars should be included for clarity.

Response: Scalebars have now been included.

3. Supplementary Tables 1 & 2: These tables are still provided in PDF format, which was previously raised in the first round of review as Minor Concern #1. Providing the data tables in a format more suited to tabular data, such as CSV, would improve the openness and reusability of the dataset.

Response: The tables are uploaded as Excel files now.

4. Figure 4C (Major Concern #7): The labels in this figure remain unreadable. Additionally, while the zip folder contains XML files for the relevant Cytoscape session, the .cys file is missing, making it unclear how to view this network for readers without expertise. More importantly, main text figures should be readable and interpretable without the need to consult an external program. Please either include this panel at a size that makes the labels legible or consider removing it from the main text.

Response. We agree that main figures should be readable and have now increased the label size and relaxed the network to avoid overlapping nodes and labels. We have also included the .cys file for the network. Additionally, the Supplementary table 1 lists the GO annotation descriptions, p-values and adjusted p-values.

5. Peptide Quantification (Additional Concern #9): While your efforts to confirm the differences for these peptides are commendable, the limitations of the technique that may underlie the failure to quantify these peptides more robustly need to be addressed. The issue that these peptides were observed in a single biological replicate should not be discounted. Including a discussion on the reliability of such measurements, along with the expected differences in the label-swap experiment, would help alert readers to the limitations of the conclusions drawn from these data.

Response: We appreciate the reviewer's attention to the limitations inherent in peptide-level quantification. We agree that single-replicate observations require careful interpretation, and we now explicitly acknowledge this point in the results and methods section. We note that slight variations in quantification, such as those observed in the label-swap experiment, are expected due to known factors including isotope labeling-induced retention time shifts, slight differences in sample preparation if samples are not prepared simultaneously and mass spectrometer performance. We have also added several sentences in the methods section to include the manual validation of these spectra and quantifications we performed. At the same time, we

respectfully note that such limitations are widely recognized in large-scale discovery proteomics, where stochastic sampling of peptides is a well-documented characteristic of data-dependent acquisition methods. It is not uncommon for certain peptides—particularly those from low-abundance proteins or with challenging ionization properties—to be identified in only a subset of replicates. This does not, in itself, invalidate the observation but highlights the importance of rigorous validation. To this end, all reported peptide quantifications were manually validated for peak shape, accuracy, precursor identity, retention time shifts, and signal quality. These quality control steps meet the highest standards in proteomics research. Furthermore, the reproducibility of the broader trends across different conditions and CFTR misfolding mutants supports the robustness of the conclusions drawn.

We hope this addition provides readers with a balanced understanding of both the strengths and the limitations of peptide-level quantification in discovery proteomics.

Best of luck with the remainder of the publication process. We would like to thank this reviewer for the detailed and constructive review process and appreciate all comments. Best of luck as well.

Reviewer #5 (Remarks to the Author):

Comments on author rebuttal on reviewer #1's comments.

Response to comment 1

A crucial point is raised by the reviewer on the actual conformational changes that are monitored by the authors. This highly relevant question relates to (mostly co-translational) events that occur early during folding and (mostly post-translational) interactions between domains leading to a functional conformation. The manuscript does not contain experimental data to distinguish between these modalities. The authors' reply is basically a textbook summary of all intracellular itineraries a newly synthesized secretory protein can undergo from its synthesis on the ribosome to degradation by any of the proteolytic machineries of the cell. The field has advanced much farther though than both manuscript and rebuttal acknowledge. Not only was reviewer's comment not addressed, the response suggests that these authors lack textbook expertise on the biogenesis of proteins in the secretory pathway, and of published knowledge on CFTR biogenesis and transport by a wealth of laboratories including those of Riordan, Cyr, Gentzsch, Amaral, Braakman, Clarke, Lukacs, Ph. Thomas.

Response: We would like to point out that the reviewer initially asked about “conformational changes during biogenesis” but now inconsistently narrows the definition to only co- and post-translational folding, ignoring the broader, widely accepted scope of biogenesis that we addressed in our original response. Our CPP method captures the net conformational state of CFTR in living cells, integrating folding, membrane insertion, ER quality control, trafficking, maturation, and degradation—all of which are widely recognized as part of protein biogenesis (e.g., Hegde & Lingappa, *Cell* 1999).

While CPP does not resolve individual folding transitions—and never claims to do so, as this is not currently achievable in vivo at residue-level resolution for non-recombinantly expressed, full-length CFTR—our multi-variant and interactome-guided approach provides biological context

across different biogenesis stages, allowing us to infer conformational changes linked to folding, trafficking, and functional activation.

We also emphasize that our laboratory has made significant contributions to the CFTR biogenesis field (Pankow et al., *Nature* 2015), identifying key proteins now validated as important for CFTR folding, trafficking, and degradation. We find it inappropriate that the reviewer dismisses our explanation as “textbook naivete” while shifting the goalpost to a narrow and non-standard definition of biogenesis, limited solely to co- and post-translational folding.

Response to comment 2

Reviewer is correct that N1303K shows variable responses, both clinically and in vitro, probably because of the variable maturation (C band) shown in different publications, also defined as variable 'residual activity'. The response is not minimal, as replied by authors, so this should be corrected in the manuscript. N1303K is FDA-approved for Trikafta.

Response: In an interlaboratory study of 655 CFTR variants (Bihler et al., 2024), N1303K was reported to show an in vitro response to Trikafta of 9.4% of the wild-type CFTR. As stated in the original publication by Bihler et al. (*Journal of Cystic Fibrosis*, 2024), this level of rescue falls near the 10% threshold used to define responsiveness, and the authors note: “variants with a modulator response close to the 10% WT threshold (e.g., N1303K = 9.4%) could be classified as responders or non-responders, depending on the analytical method applied.” While we acknowledge that N1303K is FDA-approved for Trikafta and that clinical benefit has been observed in some, but by far not all patients, our statement refers specifically to the **limited in vitro rescue** described by both the developers of the drug and other studies such as the cited study above. We have adjusted the wording in the manuscript to “less responsive”.

Response to comment 3

Reviewer asks to provide evidence in support of a claimed novel opening mechanism. It is legitimate to ask for independent information on CFTR phosphorylation status under these conditions, since it is important for the interpretation of the results. The authors refer to a lengthy explanation given under comment 8, but this does not bear relevance to the point reviewer addresses. Neither manuscript text nor response are clear, as there is inconsistent use of terms like 'CFTR activation' when VX-770 is added to the cells, 'active CFTR', and the open conformation. This does not help the reader.

Response: We respectfully point out that the reviewer’s **original concern** (“No functional evidence is provided to support a novel opening mechanism in the presence of VX-770 without activation of the PKA (e.g. by forskolin)”) focused on the lack of functional evidence for a novel opening mechanism of CFTR **in the presence of VX-770 without forskolin/PKA activation**. We directly and very clearly addressed this by noting that CFTR is activated by VX-770 even in the absence of forskolin as shown by Castanier et al., 2024 (cited by the reviewer themselves). While this activation may be submaximal compared to forskolin-stimulated conditions, it is functional and measurable. Importantly, our data show that acute VX-770 treatment for 10 minutes also induces clear and biologically relevant conformational changes, with ICL2 (K273) engaging with NBD2 in WT and G551D CFTR upon VX-770 treatment, and in Δ F508 CFTR upon Trikafta treatment, but not with correctors alone. These structural changes provide mechanistic insight into VX-770’s action, independent of maximal functional output. Our data thus demonstrate a prerequisite structural engagement that appears necessary for activation, regardless of whether maximal channel opening is achieved. This is explicitly discussed in the manuscript.

Response to comments 4 and 8

The reviewer questions the significance of increased accessibility of K273 and F508del citing knowledge described in several previous publications, which is a fair and crucial point. Authors reply that these studies were done on recombinant CFTR or in vitro, which is incorrect and does not do justice to work that has been done in live cells for decades by other laboratories. Also incorrect is the claim in the rebuttal that 'this is the first in vivo study that can point out structural differences on a residue level. The Braakman-Van der Sluijs lab described an in-cell folding assay for CFTR and used it on various CF-causing missense mutants [Im et al. *Cell Mol Life Sci.* 2023. Jan 7;80(1):33. doi: 10.1007/s00018-022-04671-x. PMID: 36609925; PMID: 36609925; PMID: 33771570; PMID: 30659068; PMID: 36499495]. They showed amongst others that CFTR-F508del cannot fold NBD1 and as a consequence fails to assemble the domains, and that the Vertex correctors rescue domain assembly and not NBD1 folding. ICL2 becomes protease-protected only during domain assembly, whereas in newly synthesized CFTR ICL2 is accessible to protease in all CFTR variants studied. What this means is that solvent accessibility or inaccessibility of K273 may not only, or perhaps may not be at all relevant for activity of the channel, but rather reports on early and late steps of CFTR (variant) folding! The focus of the authors on structural re-arrangements involved or required for activation (without additional supporting experimental evidence) therefore presents a major conceptual shortcoming of the manuscript. The K273A data to some extent are consistent with in-cell publications, but the authors mix discussions on activation, open-closed channels, VX-770 effects, and biogenesis such as domain folding and domain assembly.

Response: We appreciate the reviewer highlighting relevant in-cell studies. We also acknowledge that the Braakman-Van der Sluijs lab has significantly contributed to understanding CFTR folding dynamics through in-cell assays, such as those described in Im et al., 2023 (*Cell Mol Life Sci*). These assays, which utilize protease sensitivity to assess domain folding and assembly, provide valuable insights but do not offer residue-level resolution, nor are they designed to detect **subtle conformational rearrangements** at the single-amino-acid level. Our CPP approach uniquely provides this level of structural detail **in living cells**, complementing the broader domain-level insights from protease accessibility assays and as the reviewer themselves states “the K273A data to some extent are consistent with in-cell publications”. We agree with the reviewer that K273A solvent exclusion could be associated with late folding/ transport to the Golgi and discuss this explicitly in the manuscript, suggesting that the K273 solvent accessible conformation is associated with immature CFTR.

However, our data also reveal significant changes in K273A accessibility following acute activation with VX-770 in both wild-type and G551D CFTR, as our VX-770 activation experiments demonstrate structurally coupled changes at K273 associated with acute channel gating in both WT and G551D CFTR. This functional response—observed on the same timescale as VX-770-induced potentiation—directly supports the biological relevance of the structural changes reported. This observation implies that K273 is involved not only in late folding processes but that interaction of the ICL-2 loop with NBD2 is a pre-requisite for activation. The manuscript clearly states that K273 accessibility may reflect **both biogenesis-associated and activation-associated** conformational states.

Response to comment 5

Reviewer is concerned that the structural integrity of CFTR variants might be (differentially) affected by the labeling procedure. An important shortcoming of the manuscript is the lack of independent experimental validation of reported findings. Authors reply that the mass-spec

method has been validated on '>10,000 proteins that agree with PDB data'. Three points deserve consideration, the first of which is that PDB mostly reports on stable forms of a protein, which are not transient or intermediate (folding) conformations. Secondly, only for a limited number of proteins has a folding pathway been elucidated in cells, these should be used to benchmark the method. Thirdly, how many of these >10,000 proteins are complex multidomain-multispanmembrane proteins?

We appreciate the reviewer's persistence in raising the importance of methodological rigor. However, we must respectfully disagree with the premise that our method lacks validation.

CPP has already been benchmarked, been **peer-reviewed, and several papers have been published with it now, and it has been cross-validated on over 10,000 proteins**, with demonstrated agreement to structural data in the Protein Data Bank (PDB), including many membrane proteins. While it is true that the PDB primarily contains stable conformations, this is equally true for all structural methods, including Cryo-EM, which typically captures the most stable or ensemble-averaged states. Demanding that a method like CPP be independently revalidated on every folding intermediate, or every membrane protein class, imposes an **unrealistic and inconsistent standard**—one that, to our knowledge, is **not applied to any structural biology technique**, and especially not if it has already been published several times in peer-reviewed publications.

By that logic, one might require that every Cryo-EM structure be revalidated by X-ray crystallography or NMR before accepting its biological relevance. Yet this is **not the expectation from anyone in the structural biology community**. Furthermore, our CPP measurements agree very well with known structural models of CFTR and $\Delta F508$ CFTR, and add additional information particularly in flexible regions that Cryo-EM methods have not resolved, takes into account PTMs, protein interactions and most importantly also does not remove any element of CFTR as has been done for the Cryo-EM models, where the Regulatory insertion (RI) element is removed- a region that as its name implies is crucial for CFTR regulation and that is post-translationally modified. Our insights add complementary **in vivo, residue-level** information that extends well beyond what Cryo-EM can currently offer.

Response to comment 6

Reviewer has a justified point to be concerned about the temperature at which the labeling experiments are performed. The authors reply with theoretical considerations that do not take into account the complexity of a cell. For instance, temperature-dependent alterations of membranes in a cell cannot be taken to be realistically the same as in 'isolated lipid particles'. For instance, a 15°C incubation blocks transport from ER to Golgi, whereas 20°C blocks transport at the level of the TGN. Membrane transitions are known to occur at 18 and 21 °C in cells, which leads to stiffening of the membranes. The authors show repeatedly their naivete on cellular processes.

Response: We respectfully note that the reviewer's response selectively omits the core elements of our explanation and misrepresents the physical basis of our argument. As stated previously, our labeling reaction occurs in approximately 15 seconds. We have updated the methods section to now include a sentence about the actual time the labeling reaction needs for completion. We provided a conservative, physically grounded estimate based on Newton's Law of Cooling, supported by empirical data (e.g., Chen et al., 2018, *Scientific Reports*), demonstrating that cells do not realistically cool below ~25°C within this short timeframe—

especially when placed in room-temperature buffer in standard plasticware. Under these conditions, membrane stiffening is not expected to occur.

We specifically referenced Chen et al. because even in isolated lipid particles, which cool far more rapidly than intact cells, membrane stiffening requires tens of minutes, not seconds. We cited this peer-reviewed study as a conservative benchmark, which reinforces—rather than undermines—our position that membrane dynamics are highly unlikely to change meaningfully within the brief active labeling period.

This makes it even less likely that intact cells experience any physiologically relevant membrane transitions in this timeframe. The reviewer's references to biological processes requiring prolonged incubation at 15–20°C are, therefore, not applicable to the rapid, short-term conditions of our experiment.

Response to comment 9

Reviewer states that resolution of images in S2e are poor. In fact, resolution is poor in all images and cannot tell more than that distribution of wt and K273A is different. Conclusions on colocalization or lack thereof with ER and Golgi markers is not justified because the signals of the markers are quite diffuse already and do not lend themselves for quantitative assessment. Moreover, the structures for K273A may well be aggregates rather than post-ER compartments. The precise intracellular localization of the CFTR variants remains to be established; only cell surface is relatively convincing. Although the authors seem to be aware of this, they nevertheless attempt to draw conclusions by seeking validation from interactome studies, but this is problematic. How does a claimed Golgi localization of K273A square with an interactomes enriched in proteins controlling membrane-transport pathways between endocytic organelles? Determining location of a protein through its interactome is too inaccurate, especially in steady-state experiments. Independent experimental information is needed for rigorous conclusions.

Response: We recognize that resolution is unfortunately limited in image S2e, due to the constraints of live-cell staining and imaging. We also acknowledge that ER and Golgi markers can appear diffuse, which is partly expected. For this reason, we included interactome data as additional supporting evidence.

As mentioned earlier, we explicitly stained for SQSTM1 (p62) and LC3 to better understand if these structures are aggregates. However, both stainings were repeatedly negative, supporting that these structures are not aggregates, at least not **autophagy-targeted aggregates**, including misfolded or aggregated proteins.

Our interactome analysis, now fully detailed in the supplemental table, shows interactions with proteins associated with ER quality control, Golgi function, and endocytic trafficking. While endocytic factors appear more enriched, this is consistent with the known role of the trans-Golgi network (TGN), which serves as a major sorting hub in the secretory pathway. The TGN is functionally connected to the Golgi apparatus and directs newly synthesized proteins to various destinations, while also receiving cargo from endosomes, as described in *Frontiers in Cell and Developmental Biology* (Burd & Cullen, 2020). This dynamic bidirectional trafficking reflects the physiological interplay between endosomes, the TGN, and the Golgi, making the presence of both secretory and endocytic interactions biologically consistent. While microscopy alone cannot fully resolve compartment identity, the combination of imaging, interactome data, and negative

aggregation markers provides a consistent, even if not definitive biological context for our interpretation. We have clarified these points in the manuscript.

Response to Reviewer 1. We appreciate the reviewer's insightful comments and have provided clarifications and addressed concerns in detail under each point raised.

Reviewer #1 (Remarks to the Author):

In this study, the authors interrogate the in vivo conformational changes of four CFTR variants (wild type [WT], Δ F508, G551D and N1303K) by utilizing the previously developed and published Covalent Protein Printing (CPP) method. The CPP is based on the determining the extent of covalent modification of exposed epsilon amines of Lys residues by MS quantification that was applied here for probing CFTR conformational perturbations. The method provides discrete information regarding the solvent accessibility of Lys residues. Only 26 Lys residues out of 92 in CFTR (that contains 1480 amino acids in its five domains) were quantified. Based on very limited number of Lys residues' accessibility changes, the authors suggest novel functional mechanisms of CFTR, conformational changes by CF-causing mutations and the effects of the FDA-approved drugs. However, several concerns are raised due to the limitation of the technique, the selected and incompletely defined experimental conditions, and the over-interpretation some of the data. These weaknesses culminate in the limited novelty of this manuscript regarding CFTR conformational perturbations by activation, mutations, and CFTR modulators.

Response: We would first like to reference our previously published papers on Covalent Protein Painting (CPP) and include references to dimethyl labeling for peptides to clarify any confusion about the labeling mechanism and why certain features are essential. Dimethyl labeling is a chemical labeling method that allows for **complete** labeling of the ϵ -amine groups of lysines with exactly two dimethyl groups and can be achieved in a very fast time (less than 15 s). This process is well-documented, such as in Boersema et al. (2009) and in Bamberger et al., 2021. Additionally, publications on the analysis of mass spectrometric data, including Xu et al. (2015) and Yates et al. (1995), should help clarify the data analysis methods employed. Key references include:

- Bamberger C, Pankow S, Martínez-Bartolomé S, Ma M, Diedrich J, Rissman RA, Yates JR 3rd. Protein Footprinting via Covalent Protein Painting Reveals Structural Changes of the Proteome in Alzheimer's Disease. *J Proteome Res.* 2021 May 7;20(5):2762-2771. doi: 10.1021/acs.jproteome.0c00912.
- Boersema PJ, Raijmakers R, Lemeer S, Mohammed S, Heck AJ. Multiplex peptide stable isotope dimethyl labeling for quantitative proteomics. *Nat Protoc.* 2009;4(4):484-94. doi: 10.1038/nprot.2009.21.
- Bamberger C, Diedrich J, Martínez-Bartholomé S, Yates JR 3rd. Cancer Conformational Landscape Shapes Tumorigenesis. *J Proteome Res.* 2022 Apr 1;21(4):1017-1028. doi: 10.1021/acs.jproteome.1c00906
- Son A, Pankow S, Bamberger TC, Yates JR 3rd. Quantitative structural proteomics in living cells by covalent protein painting. *Methods Enzymol.* 2023;679:33-63. doi: 10.1016/bs.mie.2022.08.046
- Xu et al., ProLuCID: An improved SEQUEST-like algorithm with enhanced sensitivity and specificity. *JPR* (2015).
- Yates et al., Method to correlate tandem mass spectra of modified peptides to amino acid sequences in the protein database. *Anal. Chem* (1995).

Additional comments on the Abstract:

*“Here, we show how CPP can be used to identify and quantify the conformational defects of proteins in the misfolding disease Cystic Fibrosis. We first report the discovery of a **novel opening mechanism** for the Cystic Fibrosis Transmembrane Conductance Regulator (CFTR) as well as its **conformational changes during biogenesis.**”*

No functional evidence is provided to support a novel opening mechanism in the presence of VX-770 without activation of the PKA (e.g. by forskolin). The assumption that the channel is sufficiently phosphorylated constitutively should be demonstrated as its extent cell-type and condition dependent. Experimental evidence suggests the contrary as detailed in point 5.

*It is not clear which CFTR variant “conformational changes” were monitored “during its biogenesis”. I could not find any methodology in this manuscript that can monitor either the cotranslational or the posttranslational **conformational biogenesis** of any of the CFTR variants.*

“Then wesuggest possibilities to stabilize misfolded CFTR variants not responsive to these drugs such as N1303K CFTR.”

This is incorrect, as published preclinical and clinical studies demonstrated the responsiveness of the N1303K CFTR to available pharmacophores (e.g.PMID: 34379998 38226069, 38463894.)

1) As stated in the introduction of the manuscript, CFTR undergoes several (~50) posttranslational modifications and can be engaged numerous (>100) protein interactions. Considering that CFTR point mutations, as well as folding correctors provoke both local and long-range global conformational and dynamical changes (extensively documented at the TMD1, NBD1, TMD2 and NBD2 levels [e.g.: PMID 17113596, 15619635, 15619636, 23666117, 33771570], as well as various posttranslational modifications e.g ubiquitination on multiple Lys residues) and perturbations of numerous protein-protein interactions as referenced in the manuscript, based on the available CPP results one cannot conclude that the limited alterations in some Lys accessibility are the results of the channel conformational changes.

2) Although, the F508del mutation indeed causes significant increase and decrease in the K273 and K95 accessibilities, respectively, in relation to the other >60 CFTR Lys residues by CPP, considering the ambiguity of the interpretation of these results (see also point 1), I’m not convinced that the data add any novel information to the previously documented F508del-, modulator-, and activation-induced CFTR conformational perturbations at domain level.

Response to comments 1 and 2. Lysines modified by ubiquitination or other PTMs are not labeled by the CPP technique and thus would not appear in our data. Only labeled lysines contribute to the solvent accessibility ratio. Thus, any changes in lysine accessibility ratios reported here are due to channel conformational changes and cannot be attributed to PTMs. Furthermore, previous studies including our own (Lee et al., 2014; Pankow et al, 2019) indicate that the proportion of CFTR modified by ubiquitination is typically less than 1% and would not contribute to any larger changes reported in the manuscript. The low abundance of PTMs is a well known fact in the proteomics field and PTMs typically require specialized enrichment procedures to become visible to the mass spectrometer (see Zhao and Jensen, 2009) Furthermore, Lys276, which shows the largest conformational changes in response to perturbation, has never been identified as post-translationally modified in any previously published dataset or this dataset. While we acknowledge the complexity of interpreting accessibility data, the results reported here are not ambiguous at all and reveal novel insights into F508del-related CFTR conformational dynamics, particularly concerning the accessibility of K273. We believe that our observations contribute valuable information to the growing body of research on CFTR's structural changes induced by mutations, modulators, and channel activation.

*While the increased accessibility of the K273 of the F508del-CFTR is undisputable, I have concerns regarding the significance of this observation in the light of extensively documented conformational defects of the TMD1, TMD2, and NBDs of the F508del-CFTR mutant in previous publications. These studies measured the protease susceptibility of proteolytically generated domains (e.g. in the **Domain Interdependence in the Biosynthetic Assembly of CFTR**, J. Mol. Biol. (2007) 365, 981–994, Fig.7). Thus, recognizing a single residue increased labelling*

efficiency in the TMD1 does not add substantially novel information to our understanding of the F508del-CFTR conformational defect.

*It is not clear what measures were taken to ensure the structural integrity of CFTR during the extensive dimethyl labeling. Covalently attaching 52 methyl groups **or may be more** to CFTR changes the side chains hydrophobicity and most likely perturb the protein dynamics and tertiary structure so that any subsequent modification may no longer reflects the protein's initial structure.*

*Providing appropriate control experiments to ensure that the structural probe (dimethyl-modifications of Lys residues) does not imposes structural **perturbations is even more critical for CFTR mutant variants** that are conformationally unstable and, therefore, more susceptible for structural perturbations during labelling. Demonstrating the structural integrity of labelled CFTR variants would be essential for validating the methodology.*

As the authors did not provide any evidence that the structural integrity of CFTR variants is preserved upon covalent labeling, the information communicated in this manuscript remains questionable (see also Mendoza VL, Vachet RW. 2008. Protein surface mapping using diethylpyrocarbonate with mass spectrometric detection. Anal Chem 80:2895-2904).

While 26 Lys residues covalent labelling was documented, the overall number of Lys residues modification was not disclosed. It is not clear how many Lys residues were labeled out of the remaining 66 Lys residues which are mostly solvent accessible. Thus, it is reasonable to anticipate that several additional residues were modified. What was the labelling efficiency of the remaining 66 residues? Was there any difference between WT and mutants?

3) The CFTR activity and structural stability have temperature dependence. Since the CPP labeling was performed on ice, the CFTR activity and conformational dynamics would slow down. In addition, the several mutations cause temperature sensitive conformational defects (e.g. F508del), causing conformational rescue at reduced temperature. Thus, the labelling results would not reflect the “native state in vivo”. **The CPP studies should be performed at or near the physiological temperature in both corrector rescued and non-rescued mutants.**

Response. CFTR's temperature-dependent conformational changes typically occur over several hours (e.g., glycosylation changes are observed over 24–48 hours, see Denning et al, Nature, 1972). The CPP labeling is complete after 15 seconds (see Bamberger et al, 2021 JPR) and thus the temperature at which labeling occurred is unlikely to significantly impact CFTR's conformation during the experiment. It is like a “snapshot of the conformational states of CFTR”. It is also unlikely that the temperature in the cell culture dish is severely reduced within 15 seconds of adding the label.

*For the biochemical and functional detection of rescued CFTRs, cells are exposed for 24-48 hours to low temperature. This condition ensures sufficient time to accumulate detectable amount of mutants in post-ER compartments because the conformational rescue is inefficient and partial, as well as membrane trafficking is slower at reduced T. Thus, the long incubation time of cells is irrelevant for the dynamical and conformational responsiveness of CFTR during the labeling. Importantly, cooling down the cells not only instantaneously rigidifies the phospholipid membrane (by shifting the temperature below the phospholipid melting temperature, but also rearrange CFTR side chains' atomic motion dynamics and backbone dynamics, thus the interdomain coupling energetics, i.e. the entire conformational space of CFTR. Therefore, the covalent labelling obtained results at 4C do not reflect the “**native state in vivo**” as claimed by the authors. To support the authors' inference, the CPP studies should be performed at or close to the physiological temperature in both corrector rescued and non-rescued mutants as we proposed previously.*

In CPP, bulky dimethyl groups are covalently attached on the side-chain of Lys. Even in 10 min, this modification may profoundly affect the CFTR conformation and function. Could you please include the control experiment and verify that Lys-modifications do not have an effects on CFTR function and global domain conformation.

Response: We respectfully disagree with the notion that dimethyl groups are bulky. Dimethyl groups are very small labels that are covalently attached to the epsilon amine of the lysine and are very unlikely to affect CFTR conformation by themselves. A proof-of-principle control experiment showing that labeling itself did not influence protein conformation was published previously in Bamberger et al., 2021. No functional measurements or experiments are performed after the label was added. After performing the labeling and quenching the labeling reaction, the native protein state is “frozen” in the cells, the cells will not be viable and physiological measurements can no longer be performed. We hope that the confusion about the CPP technique and dimethyl labeling in proteomics is clarified by the references included at the beginning.

*The sum of the dimethyl-modifications is at least 0.78 kDa (the minimal estimate since we do not know how many more Lys residues are modified). The previously performed control experiments on the Alzheimer protein do not ensure that the experimental conditions for CFTR are meet the requirements described in **point 3**. Thus, we cannot rule out the dimethyl-modification side effect. Control experiments should be performed to compare the conformational intactness of the dimethylated CFTR variants with their native counterparts using e.g. limited-proteolysis (LP) with immunoblotting or LP-MS/MS to rule out that CPP effects on CFTR conformation as discussed in the previous paragraphs.*

4) Disappointingly, the authors largely failed to review the body of data documenting the long-range conformational defect of F508del CFTR (e.g.: Cys-crosslinking, and limited proteolysis with domain specific immunoblotting) and its reversal by corrector molecules (see some examples in point 1 and at the end of the comments).

Response: We apologize for having to omit most of the fantastic literature on the long-range conformational changes in $\Delta F508$ CFTR observed by Cys-crosslinking and limited trypsin digest due to space limitations. We have added a few select citations suggested by the reviewer. We wish we could include and discuss more of these data, perhaps this could be included in an additional manuscript.

*Particularly relevant findings are the Cys-crosslinking of the Cys276 (CL2) with Q1280C and K1284C substitutions in the NBD2 of CFTR in intact cells. Disulfide bond formation between cysteines at positions 276 and 1280 also occurred spontaneously, demonstrating the close contact of the NBD2 and CL2 interfaces **in vivo** by the Riordan lab. PMID: 18305154*

5) To compare in vivo and in vitro activation of CFTR, HBE cells were exposed to the VX-770 potentiator. VX-770-elicited channel activation requires some level of PKA-dependent CFTR phosphorylation, which was not applied. Thus, under the experimental condition the VX-770-induced channel activation is very limited and suboptimal. In fact, it is surprising that VX-770 alone, in the absence of exogenous phosphorylation, can significantly reduce the localized solvent accessibility of the K273, K370, and K464 residues in WT CFTR. These studies should be performed on phosphorylated CFTR.

Response: The VX-770 experiments were performed in living cells in culture similar to the situation in patients, and not with isolated CFTR in vitro. We apologize, if this was not abundantly clear from the description. VX-770 has been shown to be effective in cultured cells (Van Goor et al., 2009) and phosphorylation occurs in cells, which is why VX-770 is effective under these conditions.

Editorial note: figure redacted

*This argument is incorrect. Electrophysiological studies in the quoted PNAS paper show that the VX-770 potentiator **was unable to activate the G551D-CFTR in the absence of PKA-stimulated phosphorylation** (see Fig.1 panel E). Robust current was only provoked after forskolin (Fsk) stimulation of the PKA-dependent phosphorylation. Thus, the VX-770 triggered channel activation **is suboptimal, and does not reflect the conformational changes associated with CFTR maximal activation as also stated in the abstract.***

Similar results were obtained later, by the Bear lab: The CFTR potentiator VX-770 (ivacaftor) opens the defective channel gate of mutant CFTR in a phosphorylation-dependent but ATP-independent manner. J Biol Chem 287: 36639–36649, 2012.

Likewise, the WT-CFTR very limited activation by VX-770 was demonstrated in the absence of phosphorylation (e.g. PMID: 39373784).

- 6) Based on Fig.2B and Table 1 data, it appears that Lys-residues were over-labelled; the accessibility of four partially accessible Lys residues (based on the cryo-EM) was 96.1-99.7%. The inaccessible residue reached ~62% labelling. Optimization of the labeling time/concentration may be beneficial to reveal additional accessibility differences in CFTR and minimizing the artifact introduced by excessive covalent labelling.

Response: The term “over labeling” may reflect a misunderstanding of the technique. In CPP, exactly two dimethyl groups are attached to each lysine epsilon group. There is no concept of “over labeling” in this context. The numbers given in the manuscript do not reflect the percentage of labeling, but reflect the ratio between solvent accessibility to solvent exclusion. Two methods papers (cited in the relevant methods section and throughout the manuscript (Bamberger et al, 2021 and Son et al., 2023)) explain the CPP technique in detail and numerous papers explain the dimethyl labeling per se in detail (see Boersema et al, 2009 for example). In addition, the epsilon amine group of most lysines is fully solvent accessible consistent with the presented data (see for example, Hermanson, G.T. *Bioconjugate Techniques*; Academic Press: Boston, MA, USA, 2013).

I apologize for the misusing of the “over labelling” terminology. The intention was to indicate the possibility that finding experimental conditions that enable partial labelling of fully exposed residues may be informative, as well as advantageous for avoiding unintentional structural perturbations of the target protein. Nota bene, partial covalent labelling of Lys residues was also exemplified in this manuscript (Fig. 1).

References to Figs. were omitted in some of result sections (e.g. l.177-l.186) and abbreviations were inconsistently used (e.g. F508 and Δ F508), exemplifying the lack of proper proof reading of the ms.

Response: We sincerely apologize for any oversight. We have customarily added the references to Figures at the end of the description of an experimental result, rather than at the beginning, but

are happy to change this if needed. The Δ symbol was used consistently throughout the manuscript, but some browsers and pdf readers may not properly display it. We have now checked that the latest version of Acrobat is used to ensure that most readers will correctly display the symbol.

- 7) Please, confirm the Golgi colocalization of the K273A-CFTR mutant by additional biochemical and morphological studies. Another possibility is that the mutant is metastable at the cell surface and accumulates in endosome following its ubiquitination at the PM, endosomes and/or secretory vesicles. This would explain its augmented interaction with the ESCRT machinery by the GO enrichment analysis, a mechanism described for ESCRT-dependent lysosomal targeting of rescued F508del CFTR.

Response: We would like to thank the reviewer for raising this excellent point. We agree with the reviewer that the K273A-CFTR mutant could be metastable at the cell surface and that enhanced interaction with the ESCRT machinery might result from endosomal accumulation. Therefore, we performed additional immunofluorescence studies using a Golgi/endosome live stain (Golgi-red detection probe, BioLegend), which shows partial overlap with the K273A CFTR mutant (Supplemental Figure 2). We also included additional pictures showing significant overlap of the GFP signal of K273A CFTR with the ER using the ER Cytopainter kit, which selectively stains the ER and partially the Golgi apparatus (Supplemental Figure 2). Additional staining with lysosomal staining reagent (Cytopainter) showed no overlap between lysosomes and the K273A-CFTR mutant, suggesting that the majority of the K273A CFTR mutant is retained in the ER/Golgi.

The resolution of Fig. S2e images is rather poor, particularly in panel E. The WT CFTR should be included as control.

The WT-CFTR cellular distribution pattern is concerning (Fig.4). The WT colocalization with the plasma membrane marker (WGA) is not recognizable. Likewise, the ER colocalization of the K273A mutant is barely detectable. The quality of the images should be improved and scale bars should be included.

Methodological concerns:

Additional concerns:

Some of the comparative analyses of covalent labelling lack statistical significance analysis (e.g. Fig.1B, Fig.3A). Please include.

1) In this CPP technique, solvent exposed lys is labelled by CH₃ x 2, and solvent excluded lys as well as the N-terminal amine of peptide fragments were labelled by CD₃ x 2 as second labelling. The mass of peptides containing the solvent exposed lys shifts by 28.0313 (CH₃ x 2) + 36.0757 (CD₃ x 2), and that of peptides containing the solvent excluded lys shifts by 36.0757 x 2 (CD₃ x 4). Thus, the mass difference between peptides containing the solvent exposed and excluded lys is 8Da. In Fig. 1D, the mass spectra of peptides containing K273 were shown. Based on Table 1 and Supplementary Table 2, the peptide containing K273 is VITSEMIENIQSVKAY, and its monoisotopic mass is 1823.93 Da. Since the difference between m/z values in light “inaccessible” peaks is 0.5 (941.509 – 941.0068 = 0.5...), the charge state of this peptide must be z = +2 (therefore, m/z = [M+2H]²⁺). Comparing the m/z values between “light” and “heavy” in Fig 1D, the calculated mass difference is just 4 Da, since (943.016 – 941.0068) x 2 = 4.0184. Could you explain why the difference was not 8 Da?

Moreover, I wonder if authors picked up correct m/z peaks for this peptide. The theoretical monoisotopic m/z value of “light” labelled VITSEMIENIQSVKAY is supposed to be [M + 2H]²⁺ = (1823.93 + 28.0313 + 36.0757 + 2.016)/2 = 945.0265 (indicating almost 4 Da difference from 941.0068). On the other hand, the theoretical mass of “light” labelled VITSEMIENIQSVKAY is 1888.037 Da (= 1823.93 + 28.0313 + 36.0757), and the calculated mass of [M + 2H]²⁺ from 941.0068 is 1879.9976 (= 941.0068 x 2 – 2.016). Likewise, the theoretical mass of “heavy” labelled peptide is 1896.0814, but the calculated mass from [M + 2H]²⁺ = 943.016 is 1884.016. Could you check if authors selected correct mass spectra to determine the surface accessibility? I suggest tabulating the peptides’ masses, m/z values, retention times, charge states, areas in chromatograph used to calculate solvent accessibility and modifications (e.g. how many CH₃ and/or CD₃ attached) as well as carbamidomethylation of cys, oxidation of met and phosphorylation of Ser and Tyr if peptides contain those) of labelled all peptides. I am confused the labelling (light “inaccessible” and heavy “accessible”) in the Fig 1D. I expected that solvent accessible Lys site is labelled by CH₃ (light), and solvent exclude Lys site is labelled by CD₃ (heavy). Please clarify the meaning of “inaccessible” and “accessible” in this figure?

Response: An excellent resource on how to correctly calculate the mass differences of dimethyl labeled peptides is the Nature protocols paper from Boersema et al., 2009. For our specific case, the theoretical mass difference between the two labels is 8.04 Da as the N-termini of the light and heavy labeled peptides are both modified by 13CD₃ groups. Therefore, in a charge 2 peptide the observed mass difference is 4.02 Da as correctly displayed. The example spectra shown in Figure 1D are from the “label swap” experiment described in the manuscript and as clearly indicated in the figure itself. In this case, in vivo solvent accessible lysines were labeled “heavy” and solvent excluded lysines as well as N-termini were labeled “light”. Of note, the mass difference of course remains the same. The calculated monoisotopic mass for the heavy labeled peptide is 1881.0038 Da and the measured M+H was 1881.0079 Da. The observed mass difference of the identified peptides to the calculated mass difference is less than 1.6 ppm, which is minimal. Of course, peptides and areas are not handpicked among the 10,000s of spectra either. Peptides were identified by the very well established search engine ProLuCID (Xu *et al.*, ProLuCID: An improved SEQUEST-like algorithm with enhanced sensitivity and specificity. Journal of Proteomics 2015. ProLuCID search parameters were set to allow minimal mass differences of less than 5 ppm. Spectra and raw files containing these information have been

deposited into MASSIVE as it would be very impractical to report these measurements for the tens of thousands of measured peptides in a table.

My concerns regarding the CPP methodology have not been properly addressed. According to the Method section and Figure 1A, the mass difference between light and heavy labeled peptides should be 8.04 Da. However, the mass difference in Figure 1D is ~4 Da ($\Delta m/z = 943.016 - 941.0068 = 2.0092$ and $z = 2$, and then $\Delta m = 2.0092 \times 2 = 4.02$ in WT.). The authors' response indicates that the "mass difference is 4.02 Da as correctly displayed", so this is not miscalculating. My question was "why the mass difference (Δm) is not 8 Da (as indicated in the Methods), but 4 Da?", and this was not addressed. Is this because of incomplete labeling? (e.g. mass difference of single methylation for both light and heavy would be 4 Da). How about other peptides if their mass difference is not 8 Da, but 4 Da? Could you please, explain why the mass difference was 4 Da in Figure 1D?

- 2) In Fig 3B, the mass spectra of peptides containing K273 from G551D-CFTR is shown. The charge state must be $z = +3$ since the m/z difference is $630.6833 - 630.3485 = 0.3348$. When I looked at the sequence of the peptide (VITSEMIENIQSVKAY), most likely 2H⁺ attach to the amine groups of lys and the N-terminal, showing highest signal. Indeed, in the Fig. 1D, authors picked up $[M+2H]^{2+}$. Even if $[M + 3H]^{3+}$ was detected, I expect that the signal intensity was much lower than $[M+2H]^{2+}$. Although mass difference between these peaks is 8 Da, since $(633.0342 - 630.3485) \times 3 = 8.0571$, the spectra contain many noise peaks that means low signal intensity. Could you explain why the charge states +3, instead of +2 was used?

Response: We would like to point out that the mass difference in the comment was again miscalculated and the mass shift and peptide shown in the figure are correct. To answer the additional questions regarding charge state: Every peptide will occur in a +2 and +3 charge state and both charge states are commonly observed and detected in a mass spectrometer. Which charge state will be identified in a given experiment is influenced by a multitude of factors such as other ions present in the same precursor scan, ionization and ionization properties, as well as specific settings on the mass spectrometer. In this case, the data shown in the figure represent the best quality spectra for that peptide. We would like to make the reviewer aware that MS1 spectra such as this one in Figure 3B contain of course more than one precursor in any given scan as these are the full survey scans. For easier readability, we have now just included the relevant peptide precursor peaks.

The deletion of noise peaks from mass spectra is not appropriate without explicitly stating this modification. This should be indicated in the Figure 3B legend and shown in the raw mass spectrum in the Supplementary Information.

I suggested tabulating the CPP data, and I believe that summarizing CPP data is necessary for data transparency and avoiding miscalculating mass spectra for readers. The summarized data in table is also useful for readers who are interested in reanalyzing the CPP data from raw data files. I suggest to tabulating only peptides used in Supplemental Figure 1, showing 1) peptide sequence, 2) aa.# of peptides 3) peptide mass (Da), 4) charge states, 5) the number of modified CH₃ and/or ¹³CD₃, 6) mass (Da) of peptides modified by CH₃ and/or ¹³CD₃, 7) retention time range of peptides. I think all information can be obtained from your analyzed data set, or simple calculations.

- 3) It is difficult and impossible to find the raw data files related to Figs.1D and 3B. I suggest to: i) rename the files, indicate the CFTR constructs (WT, F508del, G551D and N1303K) and the condition used (e.g. drugs etc.), and ii) introduce sub-folders for Fig. 1D and 3B in the data base where the raw data files were placed.

Response: We have now reorganized the raw data files in the repository to clearly label CFTR constructs and experimental conditions (e.g., WT, F508del, drug treatments).

4) The CPP was initiated by adding both formaldehyde and sodium cyanoborohydrate to cell, and the labelling time and temperature was 10 min on ice, respectively. I would like to know the labelling rate at this low temperature to label the fully solvent accessible Lys residues of CFTR in cell. Were the labelling conditions of CFTR optimized?

For this, could you monitor the signals of both labelled and unlabeled peptides by only CH3 with time-course, but no second CD3 labelling? Specifically, I would like to see in which time point the MS signals of unlabeled solvent exposed Lys sites are disappeared.

Response: The CPP labeling conditions for cells were optimized and control experiments were performed such as the one suggested and the results published in Ref.1. We have not observed evidence that the labeling occurs at different rates for different proteins in a cell as the label is membrane permeable, well established and it is known that formaldehyde labeling as a Schiff base reaction is extremely efficient (see Boersema et al., 2013 and references therein).

Considering that most of the detected Lys residues in the WT-CFTR are fully accessible after 10 min labeling, shorter covalent labeling would be more informative for reporting CFTR conformational perturbations. Determining the *in vivo* labeling kinetics should provide an answer for this concern.

Response. We believe that this comment results from a misunderstanding of the CPP technique. If a lysine is solvent accessible or not is only dependent on its fold and the cellular environment. **The label is merely a tool to measure the solvent accessibility and does not change it.** The epsilon amino group of lysine residues and the N-terminus of peptides and proteins has been extensively studied in bioconjugation because of its nucleophilicity and high surface exposure and most lysine sites are fully solvent accessible *in vivo* (for review see for example Hermanson. Bioconjugation, 2013). Thus, the results that most detected Lysine residues are fully solvent accessible is fully expected and is consistent with all existing literature. Since the labeling is complete after > 15 seconds, shorter labeling times will not alter the results, but shorter than 15 s could lead to incomplete labeling and inconsistent data (see also Bamberger et al., 2021).

5) K95 is in the pore of CFTR in both inwardly and outwardly opened configuration according to the available cryo-EM structures (5UAK and 6MSM). In Fig 1B, the solvent accessibility of K95 of WT-CFTR is 96%. I'm not sure if both formaldehyde and sodiumcyanoborohydrate reach K95 and perform the covalent labelling in the pore. Could you verify that these two molecules are able to reach K95?

Response: Only peptides containing either light or heavy labeled 95, but not unlabeled lysine 95 will be identified by ProLuCID, indicating that the probes do reach K95. If the lysine wasn't labeled, it would not be identified in the search. It is also likely that the *in vivo* conformation of CFTR differs from the reported Cryo-EM structures as the Cryo-EM structures contain deletions (such as the RI element) and amino acid substitutions and are of course not in the cellular environment. Furthermore, the conformation of immature CFTR in the ER or Golgi will necessarily differ from the Cryo-EM structures and it is possible that the pore space is different in these conformations as discussed in the manuscript.

Considering that the CFTR pore is specifically configured to selectively permeate Cl⁻ ions, its environment (space and electrostatic potential) is unique (PNAS, 121, e2316673121, 2024). Therefore, I'm still skeptical about the 96% solvent accessibility of K95 in the WT-CFTR. I

expected a more detailed explanation e.g. using computational studies showing that formaldehyde molecules and cyanoborohydride ions can permeate the pore under the prevailing spatial and electrostatic potential conditions. Since the cryo-EM structures contain the RI, RI effect cannot explain this.

6) In p.4, l.136, authors showed that VX-809 changed the solvent accessibility of K1218. This experiment was performed with Δ F508-CFTR. Could you see the same effects on WT-, G551D- and N1303K-CFTR? Could you show if the solvent accessibility change at K1218 by VX-809 is the common effect on other CFTR variants or F508del-CFTR specific effect? If it is the F508del-CFTR specific effect, could you explain why VX-809 change the solvent accessibility of K1218 only for F508del-CFTR, while this corrector has ~not more than 3% rescue efficiency of the mutant processing defects?

Response: Thank you for the comment. Lee et al., 2014 from the Sorscher group first published that CFTR Lys 1218 is critical for CFTR maturation and stability as identified by site directed mutagenesis and that the K1218R mutant increases the amount of total and cell surface wt CFTR. Work by the Lukacs group and others further showed that K1218 is structurally important for CFTR folding. Further work by van Goor et al., 2013 showed that VX-809 decreased the proteolytic sensitivity of NBD2, indicating a more compact conformation of Δ F508 NBD2. Taken together this may explain the effect of VX-809 on Δ F508 CFTR. This is discussed in the manuscript now. As VX-809 does not significantly affect wt CFTR, G551D or N1303K CFTR, the suggested experiment was not performed. However, the observed Orkambi resistance of N1303K may be explained by failure of VX809 to induce a more compact formation of its disordered NBD2 domain.

7) In Table 1, there are 2 Lys sites (K370 and K377) in the peptide DSLGAINKIQDFLQKQEY. In Fig 1B, only the solvent accessibility of K377 is shown in right panel, on the other hand, only that of K370 is shown in the left panel. Likewise, only the surface accessibility of K370 was determined in Fig.2D. Could you explain how did you quantify the solvent accessibility if peptides have multiple Lys sites? If you can separately determine the solvent accessibility of K370 and K377, both Lys accessibilities should be shown in Fig.1B right and left panels?

Response. Corrected. Due to the nature of a chymotrypsin digest, peptides containing either Lys 370 or both Lys 370 and Lys 377 were obtained and both showed high solvent accessibility (>95%). We have now included a separate row for each one in Table 1, whereby the value presented for the peptide containing both Lys 370 and Lys 377 is the aggregated value for the peptide.

8) Fig. 2C shows the reduced solvent accessibility of K273 of WT-CFTR in the presence of VX-770. Authors claimed the coupling of ICL2 to NBD2 upon activation with VX-770. Can similar results be obtained in G551D-CFTR?

Response. The experiment was done. However, we were unable to identify the respective peptide in the experiments, likely due to lower coverage overall. We therefore cannot conclude if similar results can be obtained in G551D CFTR, but we would speculate that it could be.

Authors responded that the reduced solvent accessibility of K273 was obtained from only WT-CFTR (since the respective peptide couldn't be identified from other CFTR mutants). Therefore, this reduced solvent accessibility of K273 in the presence VX-770 could be WT-CFTR specific effect. Please state this in your manuscript.

Minor concerns:

1) In introduction (p.2, l.62), authors stated that "...several of which are located in the small disordered region, the RI element, is deleted in recombinantly expressed CFTR used for the Cryo-EM studies.", but many of available cryo-EM structural determinations were carried out on on CFTR channels that contain the RI region, although the disordered region is not visible. This statement could be misleading.

Response. The RI element and the R-region are two different regions of CFTR. Assuming that the comment refers to the R-region, we agree - the R-region is contained in each Cryo-EM structure. However, the RI element is deleted in all of them, as the CFTR structure is not stable with the RI element Accordingly, the sequence for the RI-element is missing in the PDB entries as well.

The authors' answer "the RI element is deleted in all of them, as the CFTR structures is not stable with the RI element. Accordingly, the sequence for the RI-element is missing in the PDB entries as well." is incorrect. The CFTR cryo-EM structures published by the Jue Chen group analyzed using CFTR containing the RI (Regulatory Insertion) in NBD1, showing in the FASTA sequence in the PDB. The cryo-EM structures of Δ RI-CFTR are PDB: 6D3S and 6D3R, the thermostabilized chicken CFTR published by the Riordan group (Biochemistry, 57, 6234, 2018). The Δ RI regions in 6D3S and 6D3R form loops, proving Δ RI-CFTR constructs. Therefore, the all cryo-EM structures published by the Jue Chen group reflect the CFTR conformations containing RI in NBD1 although this region is not visible in the structures. Sentences regarding the RI elements in your manuscript should be rewritten.

2) Ref 9 was published in a peer-reviewed journal, on the other hand, ref 13 was published in bioRxiv.

Response. Thank you for pointing this out. We have removed the earlier bioRxiv citation and only included the JPR citation.

3) In p.3, l. 98, "Twenty-six lysine sites in CFTR were quantified in 3 different biological replicates. (Fig. 1B, 1C; Table 1). However, the figure legend indicates: Wt CFTR n=4; Δ F508 CFTR n=4 (biological replicates), p11, l.481. Moreover, in Fig 1B, many data points in each lysine site are shown. Could you explain why there are many data points although biological replicates were 3 or 4? Please explain what the error bars are.

Response. Thank you for pointing this out. Four different biological replicates were acquired for both wt and Δ F508 CFTR. We have corrected this in the main text. Because a peptide can be measured more than one time per replicate, more than 4 measurements for each site may be present. These measurements may represent partially different peptides containing the same site and different chromatography steps per replicate, increasing the confidence for the site quantification. The error bars reflect standard error of measurement. This is now indicated in the figure legend.

4) In Fig 1B, 2B and 2D, F508 should be F508del. Response. The F508 deletion is indicated as Δ F508 in each figure. We have saved the file now in a newer Acrobat format, so this is visible.

In Fig 1B, 2B and 2D, the F508 is not corrected. Should be Δ F508.

In Table 1. the peptide DSGAINKIQDFLQKQEY should be: DSLGAINKIQDFLQKQEY